# ✂ Cutting the Skip: Training Residual-Free Transformers

**Yiping Ji**[1,2*]   **James Martens**   **Jianqiao Zheng**[1]   **Ziqin Zhou**[1]   **Peyman Moghadam**[2]
**Xinyu Zhang**[3]   **Hemanth Saratchandran**[1]   **Simon Lucey**[1]

[1]Australian Institute for Machine Learning, Adelaide University
[2]DATA61, CSIRO
[3]University of Auckland

## Abstract

Transformers have achieved remarkable success across a wide range of applications, a feat often attributed to their scalability. Yet training them without residual (skip) connections remains notoriously difficult. While skips stabilize optimization, they also disrupt the hierarchical structure of representations, raising the long-standing question of whether transformers can be trained efficiently *without* them. In this work, we address this problem by analyzing the Jacobian of a skipless transformer block, showing why residuals improve conditioning and revealing that their stabilization benefits can be recovered through a principled initialization strategy. Building on this insight, we introduce the first method that enables stable and efficient training of *skipless transformers* without altering the standard architecture. We validate our approach on Vision Transformers (ViTs) in both supervised and self-supervised settings, demonstrating that skipless ViTs trained with our initialization overcome the usual optimization barriers, learn richer hierarchical representations, and outperform strong residual baselines on dense prediction benchmarks. These results show that skip connections are not a fundamental requirement for training ViTs and open new avenues for hierarchical representation learning in vision models.

## 1 Introduction

Over the past decade, large transformer-based models have achieved remarkable success, demonstrating strong zero-shot and generalization capabilities across tasks and domains through a single, reusable model (Caron et al., 2021; Comanici et al., 2025; Wang et al., 2025). Their ability to be trained at great depth relies heavily on skip connections (He et al., 2016), which have become a cornerstone of modern deep learning models. This unprecedented scalability in depth is widely regarded as a key factor behind the astonishing performance of transformer-based architectures.

However, the reliance on skip connections raises an important question: do such networks truly operate at the depth implied by their architecture? Prior work (Veit et al., 2016; Gromov et al., 2025) suggests that residual connections make networks behave as if they are much shallower than their nominal depth. An earlier study (He et al., 2023) was the first to investigate "skipless transformers", introducing a modified self-attention block to preserve well-behaved forward kernels. Although this modification improved trainability, the resulting models still converged significantly more slowly than their residual counterparts. This paper addresses this gap by introducing a theoretically grounded initialization scheme that does not require architectural changes. Combined with a second-order optimizer (Vyas et al., 2025), our approach enables skipless Vision Transformers to achieve training speeds comparable to residual-based models.

The concept of skip connections dates back to the 1960s: Rosenblatt et al. (1962) described a three-layer multilayer perceptron referred to as a cross-coupled system, where skip-like couplings were

---

*Corresponding e-mail:yiping.ji@adelaide.edu.au

already present. Decades later, skip connections were popularized in ResNets (He et al., 2016) and subsequently adopted in transformers (Vaswani et al., 2017), and they are now considered crucial for training very deep networks. One proposed explanation for their effectiveness is that they improve the conditioning of the network Jacobian, thereby facilitating gradient flow and enabling faster, more stable convergence (Ji et al., 2025b). Empirical evidence also suggests that self-attention tends to be disproportionately ill-conditioned—acting as an optimization bottleneck—compared to other components such as feed-forward networks, underscoring the stabilizing role that residual connections can play within transformers.

While skip connections are vital for optimizing modern neural networks, they also change how architectural depth is functionally expressed. Deep networks are intended to form compositional hierarchies in which representations become progressively more abstract layer by layer (LeCun et al., 2015). However, skip connections disrupt this hierarchy by continually reintroducing information from earlier layers into later ones. This shortcutting interrupts the intended progression of abstraction (Zhang et al., 2024) and can limit the network's ability to learn rich, deeply composed features. As a result, networks with skips often behave as if they are effectively shallower than their nominal depth suggests. Prior studies have shown that in ResNets, skip connections reduce the role of deep compositions, making networks behave like ensembles of shallower subnetworks (Veit et al., 2016). In modern transformers, this effect is even more pronounced: after convergence, many deeper layers contribute so little to the final prediction that they can be pruned with minimal loss (Gromov et al., 2025). Together, these findings suggest that while skip connections are indispensable for optimization, they may obscure the true representational benefits of depth — motivating our goal of designing transformers without shortcuts.

To the best of our knowledge, the only prior work to train skipless transformers is that of He et al. (2023), who modified the self-attention block to maintain well-behaved forward kernels and prevent the kernel matrix from collapsing toward rank 1. Although their method successfully removes residual connections, it does so by altering the standard transformer architecture, and the modified attention blocks are not compatible with widely used optimizations such as Flash Attention (Dao, 2024). In contrast, our approach requires *no architectural changes*: we retain the standard transformer block design and achieve stable training of skipless transformers solely through a principled initialization strategy.

Guided by the first principle of gradient-based optimization—good network conditioning (see Section 4.1)—we analyze the Jacobian of transformers and use this insight to design a principled initialization strategy that enables stable training of skipless models. Our main contributions are:

- **Jacobian analysis:** We provide a theoretical study of the transformer Jacobian and show that skip connections stabilize optimization by improving its conditioning.

- **Initialization without architectural changes:** Guided by this analysis, we introduce a simple, theoretically grounded initialization scheme that requires *no* changes to the transformer block, remains fully compatible with FlashAttention, and enables stable end-to-end training of skipless transformers.

- **Supervised training at parity with residuals:** On image classification benchmarks, skipless models trained with a second-order optimizer converge as quickly as standard residual transformers and achieve comparable accuracy.

- **Improved self-supervised representations:** In self-supervised learning, skipless models outperform residual transformers in dense prediction tasks, while being parameter-efficient, training faster and producing more semantically coherent representations.

- **Enabling depth studies:** Our approach makes it possible— for the first time— to systematically study *truly deep* (skipless) Vision Transformers, offering new insights into hierarchical representation learning in vision.

## 2 TRANSFORMERS: TERMINOLOGY AND NOTATION

A standard transformer begins with a token embedding $\mathbf{X}_0 \in \mathbb{R}^{n \times d}$, where $n$ is the number of tokens and $d$ is the embedding dimension. This embedding is then passed through a stack of $L$ transformer blocks. Each block consists of two main components: a *Self-Attention Block* (SAB)

and a *Feed-Forward Network* (FFN), as defined in Eqs. 1 and 2, respectively. The SAB applies Self-Attention (SA) together with a residual (skip) connection, while the FFN applies a multilayer perceptron (MLP), also with a residual connection. We denote by $\mathbf{X}_\ell$ the output embedding after the $\ell$-th transformer block. In summary, we have

$$\mathbf{X}_\ell = \hat{\mathbf{X}}_{\ell-1} + \text{SA}(\hat{\mathbf{X}}_{\ell-1}) \quad \text{and} \tag{1}$$

$$\hat{\mathbf{X}}_\ell = \mathbf{X}_l + \text{MLP}(\mathbf{X}_l), \tag{2}$$

Self-attention allows the network to selectively attend to relevant parts of the input and is core component of modern transformers.

Omitting $\ell$ for clarity, the self-attention operation is defined as

$$\text{SA}(\mathbf{X}) = \mathbf{AVW}^O, \tag{3}$$

where $\mathbf{Q} = \mathbf{XW}^Q, \mathbf{K} = \mathbf{XW}^K, \mathbf{V} = \mathbf{XW}^V$, and the attention matrix is $\mathbf{A} = \eta(\mathbf{QK}^\top)$. Here, $\mathbf{Q}, \mathbf{K}, \mathbf{V}$ are the *queries*, *keys*, and *values*, respectively. The parameter matrices $\mathbf{W}^Q, \mathbf{W}^K, \mathbf{W}^V, \mathbf{W}^O \in \mathbb{R}^{d \times d}$ are learnable, and $\eta(\cdot)$ is typically the softmax function.

In practice, *multi-head attention* is used. The projection matrices are divided across $h$ heads, such that

$$\mathbf{W}_i^Q, \ \mathbf{W}_i^K, \ \mathbf{W}_i^V \in \mathbb{R}^{d \times d_h}, \qquad d_h = \tfrac{d}{h}.$$

For head $i$, we compute

$$\mathbf{A}_i = \eta(\mathbf{Q}_i \mathbf{K}_i^\top),$$

and the final output is obtained by concatenating across heads:

$$\text{SA}(\mathbf{X}) = \text{Concat}(\mathbf{A}_1 \mathbf{V}_1, \ldots, \mathbf{A}_h \mathbf{V}_h)\mathbf{W}^O. \tag{4}$$

## 3 RELATED WORK ON SKIPLESS ARCHITECTURES

Many works have successfully removed skip connections in CNN architectures, overcoming optimization challenges and achieving competitive performance (Zhang et al., 2022; Zagoruyko & Komodakis, 2017; Martens et al., 2021). In contrast, in the transformer domain, to the best of our knowledge, only one paper has investigated training skipless language transformers (He et al., 2023) by modifying the Self-Attention Block. Based on the observation that skipless transformers are suffering from rank collapse (Noci et al., 2022), where the kernel matrix converges in depth to have rank 1, they modified the self-attention block to maintain well-behaved kernels at initialization. Our work differs from this previous attempt in that we focus on the conditioning of the network Jacobian instead of the properties of the kernel, our modifications are purely to the initialization of the weight matrices, and our experiments consider vision models instead of text.

## 4 NETWORK JACOBIAN ANALYSIS

### 4.1 PRELIMINARIES

Throughout this paper, when analysing the network Jacobian, we denote the transformer network as $f(\mathbf{x}; \theta) \in \mathbb{R}^{nd}$, where $\mathbf{x} = \text{vec}[\mathbf{X}]$ is the vectorized token embedding, $n$ is the number of tokens, $d$ is the feature dimension, and $\theta$ denotes all learnable network parameters such that $p = \dim(\theta)$. Importantly, $f(\mathbf{x}; \theta)$ *deliberately omits the token embedding and output-head* so that we can focus on the internal interactions of the transformer blocks; for this reason the network output is of size $nd$.

For a batch of $m$ input examples, we define the stacked output

$$F(\theta) := \big[ f(\mathbf{x}_1; \theta); \cdots ; f(\mathbf{x}_m; \theta) \big] \in \mathbb{R}^{mnd}. \tag{5}$$

The network Jacobian is then $\mathbf{J} = \frac{\partial F}{\partial \theta} \in \mathbb{R}^{mnd \times p}$, and its conditioning provides a key indicator of the network's training dynamics. We define the condition number as the ratio of the largest to smallest singular value $\kappa(\mathbf{J}) = s_{\max} \cdot s_{\min}^{-1}$.

Prior research has shown that improved transformer conditioning leads to more stable training and stronger results. For example, Ji et al. (2025a) improved the conditioning of low-rank matrices using sinusoidal activations, thereby enhancing low-rank learning without additional parameters. Similarly, Saratchandran & Lucey (2025) introduced conditioned embedded tokens, strengthening conditioning with minimal overhead. More recently, Ji et al. (2025b) argued that the primary role of skip connections—particularly within the self-attention block—is to improve conditioning, and demonstrated that transformers fail to train in their absence.

A central hypothesis of this work is that residual (skip) connections, while essential for optimization, violate the hierarchical principle of deep networks by continually injecting shallow features into deeper layers. Removing these shortcuts makes training challenging because the Jacobian of skipless transformers is poorly conditioned at random initialization (Ji et al., 2025b). Building on a theoretical analysis of the network Jacobian, we propose a principled initialization strategy that directly improves conditioning. This enables training *completely skipless transformers* at speeds comparable to standard residual models while learning richer, more semantically coherent internal representations.

### 4.2 DECOMPOSITION OF THE NETWORK JACOBIAN

The Jacobian of the transformer network $F$ can be decomposed into block columns:

$$\mathbf{J} = \frac{\partial F}{\partial \theta} = \left[ \mathbf{J}_1, \hat{\mathbf{J}}_1, \ldots, \mathbf{J}_L, \hat{\mathbf{J}}_L \right],$$

where

$$\mathbf{J}_\ell = \frac{\partial F}{\partial \theta^{(\ell)}} \in \mathbb{R}^{mnd \times p_\ell}, \qquad \hat{\mathbf{J}}_\ell = \frac{\partial F}{\partial \hat{\theta}^{(\ell)}} \in \mathbb{R}^{mnd \times \hat{p}_\ell}.$$

Here, $\mathbf{J}_\ell$ and $\hat{\mathbf{J}}_\ell$ are the Jacobians of the final output with respect to the parameters of the $\ell$-th SAB and FFN sub-blocks, respectively, and $\sum_{\ell=1}^{L} \left( p_\ell + \hat{p}_\ell \right) = \dim(\theta) = p$.

Following Ji et al. (2025b), we adopt the simplifying assumption that the conditioning of the full Jacobian is controlled by its worst-conditioned sub-blocks. In particular we write

$$\kappa(\mathbf{J}) \leq \max_\ell \left\{ \kappa(\mathbf{J}_\ell), \ \kappa(\hat{\mathbf{J}}_\ell) \right\}. \tag{6}$$

That is, the spectral condition number of the entire network Jacobian is assumed to be no larger than the worst condition number among all SAB and FFN sub-block Jacobians.

This assumption does not hold universally, but we provide justification for it under mild block-incoherence conditions with balanced blocks (see Section A.4.4). Ji et al. demonstrated both theoretically and empirically that SAB sub-block Jacobians are significantly less well-conditioned than their FFN counterparts. For this reason our focus on this paper is around the condition of the SAB sub-block Jacobian $\mathbf{J}_\ell$.

### 4.3 DIVING INTO SUB-BLOCKS: SKIP CONNECTIONS IMPROVE CONDITIONING

With skip connections, the vectorized SAB and FFN sub-block updates at layer $\ell$ are

$$\mathbf{x}^{(\ell)} = f_{\text{SA}}(\hat{\mathbf{x}}^{(\ell-1)}; \theta^{(\ell)}) + \hat{\mathbf{x}}^{(\ell-1)} \in \mathbb{R}^{nd}, \qquad \hat{\mathbf{x}}^{(\ell)} = f_{\text{MLP}}(\mathbf{x}^{(\ell)}; \hat{\theta}^{(\ell)}) + \mathbf{x}^{(\ell)} \in \mathbb{R}^{nd}.$$

Denoting the derivative of the SA and MLP output with respect to the corresponding inputs by

$$\mathbf{K}_\ell = \frac{\partial f_{\text{SA}}(\hat{\mathbf{x}}^{(\ell-1)}; \theta^{(\ell)})}{\partial \hat{\mathbf{x}}^{(\ell-1)}} \in \mathbb{R}^{nd \times nd}, \qquad \hat{\mathbf{K}}_\ell = \frac{\partial f_{\text{MLP}}(\mathbf{x}^{(\ell)}; \hat{\theta}^{(\ell)})}{\partial \mathbf{x}^{(\ell)}} \in \mathbb{R}^{nd \times nd}, \tag{7}$$

we have the derivative of the network output with respect to the SA parameters at layer $\ell$ is:

$$\frac{\partial f(\mathbf{x}; \theta)}{\partial \theta^{(\ell)}} = \prod_{i=L}^{\ell+1} \left\{ \left( \hat{\mathbf{K}}_i + \mathbf{I}_{nd} \right) \left( \mathbf{K}_i + \mathbf{I}_{nd} \right) \right\} \left( \hat{\mathbf{K}}_\ell + \mathbf{I}_{nd} \right) \frac{\partial f_{\text{SA}}(\hat{\mathbf{x}}^{(\ell-1)}; \theta^{(\ell)})}{\partial \theta^{(\ell)}} \in \mathbb{R}^{nd \times p_\ell}. \tag{8}$$

If skip connections are not present, we have:

$$\frac{\partial f(\mathbf{x}; \theta)}{\partial \theta^{(\ell)}} = \prod_{i=L}^{\ell+1} \left( \hat{\mathbf{K}}_i \mathbf{K}_i \right) \hat{\mathbf{K}}_\ell \frac{\partial f_{\text{SA}}(\hat{\mathbf{x}}^{(\ell-1)}; \theta^{(\ell)})}{\partial \theta^{(\ell)}} \in \mathbb{R}^{nd \times p_\ell}. \tag{9}$$

From Eq. (5), the Jacobian $\mathbf{J}_\ell \in \mathbb{R}^{mnd \times p_\ell}$ is the concatenation of $\frac{\partial f(\mathbf{x};\theta)}{\partial \theta^{(\ell)}} \in \mathbb{R}^{nd \times p_\ell}$ for $m$ samples. By assumption i), we have $\kappa(\mathbf{J}_\ell)$ is bounded by the largest $\kappa(\frac{\partial f(\mathbf{x};\theta)}{\partial \theta^{(\ell)}})$ for $m$ samples. Thus, comparing Eqs. (8) and (9), we can see clearly how the skip connections ($\mathbf{I}_{nd}$ term) help network conditioning. As we stated before in assumption ii), compared to the conditioning of the MLP $\hat{\mathbf{K}}_\ell$, the conditioning of the SA $\mathbf{K}_\ell$ is much worse (explored in Section 5.1 under default truncated normal initialization with Proposition 1). The addition of the identity matrix $\mathbf{I}_{nd}$ in Eq. (8) shifts the spectrum of $\mathbf{K}_\ell$ from zero, regularizing the smallest singular values.

These observations invite the question: is there an alternative way to maintain the good conditioning of $\mathbf{K}_\ell$ such that $\kappa(\mathbf{K}_\ell) \approx \kappa(\mathbf{K}_\ell + \mathbf{I}_{nd})$ in a skipless transformer?

## 5    A New Initialization to Enable Skipless Transformers

Based on the previous analysis, our goal is to improve the conditioning of $\kappa(\mathbf{K}_\ell)$. To this end, we first give an expression $\mathbf{K}_\ell$ at layer $\ell$:

$$\mathbf{K}_\ell = (\hat{\mathbf{X}}_{\ell-1}\mathbf{W}_\ell^{\mathrm{V}}\mathbf{W}_\ell^{\mathrm{O}} \otimes \mathbf{I}_n)^\top \mathbf{A}_\ell' + (\mathbf{W}_\ell^{\mathrm{V}}\mathbf{W}_\ell^{\mathrm{O}})^\top \otimes \mathbf{A}_\ell, \tag{10}$$

where $\mathbf{A}_\ell \in \mathbb{R}^{n \times n}$ is the attention matrix, and $\mathbf{A}_\ell' \in \mathbb{R}^{n^2 \times nd}$ is the derivative of the attention matrix with respect to the input $\hat{\mathbf{x}}^{(\ell-1)} = \mathrm{vec}(\hat{\mathbf{X}}_{\ell-1})$ (vectorized when forming the derivative matrix) [1].

Using this expression we will proceed to derive a principled initialization for the weight matrices $\mathbf{W}_\ell^{\mathrm{Q}}, \mathbf{W}_\ell^{\mathrm{K}}, \mathbf{W}_\ell^{\mathrm{V}}$ and, $\mathbf{W}_\ell^{\mathrm{O}}$ to improve the conditioning of $\mathbf{K}_\ell$. $\mathbf{W}_\ell^{\mathrm{V}}\mathbf{W}_\ell^{\mathrm{O}}$ appears in both term and $\mathbf{W}_\ell^{\mathrm{Q}}\mathbf{W}_\ell^{\mathrm{K}}$ appears in the $\mathbf{A}_\ell$ and $\mathbf{A}_\ell'$.

### 5.1    Initialization for $\mathbf{W}_\ell^{\mathrm{V}}\mathbf{W}_\ell^{\mathrm{O}}$

A key observation (see Eq. 10) is that the product $\mathbf{W}_\ell^{\mathrm{V}}\mathbf{W}_\ell^{\mathrm{O}}$ appears in both terms of the Jacobian. For training to be stable, this product must be well-conditioned in order to improve the condition of $\mathbf{K}_\ell$. The best-case scenario is when it is a (scaled) orthonormal matrix, because in that case all of its singular values are equal, so that $\kappa(\mathbf{W}_\ell^{V}\mathbf{W}_\ell^{O}) = 1$. To achieve this, we first initialize a random square matrix $\mathbf{Q} \in \mathbb{R}^{d \times d}$ with zero-mean, unit-variance entries. Then we perform an SVD decomposition such that $\mathbf{Q} = \mathbf{U}\mathbf{S}\mathbf{V}^\top$ and we assign $\mathbf{W}_\ell^{\mathrm{V}} = c \cdot \mathbf{U}$ and $\mathbf{W}_\ell^{\mathrm{O}} = c \cdot \mathbf{V}^\top$, where $c$ is a scaling constant. This ensures the matrix $\mathbf{W}_\ell^{\mathrm{V}}\mathbf{W}_\ell^{\mathrm{O}}$ is scaled orthonormal.

### 5.2    Initialization for $\mathbf{W}_\ell^{\mathrm{Q}}\mathbf{W}_\ell^{\mathrm{K}^\top}$

Recall the attention $\mathbf{A}_\ell = \mathrm{softmax}(\mathbf{M}_\ell)$, where $\mathbf{M}_\ell = \hat{\mathbf{X}}_{\ell-1}\mathbf{W}_\ell^{\mathrm{Q}}\mathbf{W}_\ell^{\mathrm{K}^\top}\hat{\mathbf{X}}_{\ell-1}^\top$. The conditioning of $\mathbf{A}_\ell$ critically depends on the structure of its logits $\mathbf{M}_\ell$.

**Proposition 1** (Softmax conditioning: diagonal dominance vs. diffuse rows). *Let $S_\tau(\mathbf{M}_\ell) \in \mathbb{R}^{n \times n}$ denote the row-wise softmax with temperature $\tau > 0$.*

***(Diffuse rows).***  *If each row of $\mathbf{M}_\ell$ has a small range (difference between maximum and minimum)$\Delta \ll \tau$, then $S_\tau(\mathbf{M}_\ell)$ is close to the uniform matrix $\frac{1}{n}\mathbf{1}\mathbf{1}^\top$, which has rank 1. In this case $\kappa(S_\tau(\mathbf{M}_\ell)) \gtrsim \frac{\tau}{\Delta}$, and the conditioning worsens as $n$ grows.*

***(Diagonal dominance).***  *If $\mathbf{M}_\ell$ is diagonal dominant, i.e. $\mathbf{M}_{ii} - \max_{j \neq i} \mathbf{M}_{ij} \geq \gamma > 0$, then $S_\tau(\mathbf{M}_\ell)$ is close to an identity matrix, and*

$$\kappa(S_\tau(\mathbf{M}_\ell)) \leq \frac{1 + \varepsilon(\gamma/\tau)}{1 - \varepsilon(\gamma/\tau)},$$

*with $\varepsilon(\gamma/\tau) \to 0$ as $\gamma/\tau \to \infty$. Hence $S_\tau(\mathbf{M}_\ell)$ is well-conditioned when diagonal logits are dominant.*

---

[1] $\ell$ is defined as head index in the previous section but in the following sections we redefine the $\ell$ as the block index

An illustration of this proposition is in Section A.4.1. This proposition highlights the key insight: at random initialization, the logits $\mathbf{M}$ are "diffuse", hence, the attention matrix $\mathbf{A}$ is close to the uniform matrix and thus ill-conditioned (see Section A.4.1), which is the main cause of the ill-conditioned $\kappa(\mathbf{K}_\ell)$.

To address this, we initialize the query and key projections $\mathbf{W}_\ell^Q$ and $\mathbf{W}_\ell^K$ such that

$$\mathbf{W}_\ell^Q \mathbf{W}_\ell^{K\top} = \alpha \mathbf{Z} + \beta \mathbf{I}, \tag{11}$$

where the entries of $\mathbf{Z}$ are sampled as $\mathbf{Z}_{ij} \sim \mathcal{N}\left(0, \frac{1}{d}\right)$, $d$ is the weight dimension, $\mathbf{I}$ is the identity matrix, and $\alpha, \beta$ are scalar constants. This scheme—sometimes called *mimetic initialization* (Trockman & Kolter, 2023)—has been shown empirically to improve both convergence and final performance in transformers.

Our contribution is to provide a theoretical motivation: the identity term $\beta \mathbf{I}$ encourages diagonal dominance in $\mathbf{W}_\ell^Q \mathbf{W}_\ell^{K\top}$, which in turn helps ensure that the initial attention operator is well-conditioned at the start of training. However, we emphasize that diagonal dominance of $\mathbf{W}_\ell^Q \mathbf{W}_\ell^{K\top}$ *does not automatically imply* that the transformed matrix $\mathbf{X}^\top \mathbf{W}_\ell^Q \mathbf{W}_\ell^{K\top} \mathbf{X}$ is also diagonally dominant. In Section A.4.1 we detail the conditions under which this property carries over after projection by the token embeddings $\mathbf{X}$.

A scaled orthonormal $\mathbf{W}_\ell^V \mathbf{W}_\ell^O$ and diagonal dominant attention map $\mathbf{A}_\ell$ guarantees that the second term of $\mathbf{K}_\ell$, namely $(\mathbf{W}_\ell^V \mathbf{W}_\ell^O)^\top \otimes \mathbf{A}_\ell$, is well-conditioned. The remaining question is whether this also ensures a well-conditioned $\mathbf{K}_\ell$ overall.

**Proposition 2.** *(Conditioning of $\mathbf{K}_\ell$) Let $\mathbf{K}_\ell = \mathbf{B}_\ell + \mathbf{E}_\ell$, where $\mathbf{E}_\ell = (\hat{\mathbf{X}}_{\ell-1} \mathbf{W}_\ell^V \mathbf{W}_\ell^O \otimes \mathbf{I}_n)^\top \mathbf{A}_\ell'$ (the "**perturbation term**"), and $\mathbf{B}_\ell = \mathbf{W}_l^{O\top} \mathbf{W}_l^{V\top} \otimes \mathbf{A}_\ell$ (the "**dominant term**"). With above initialization (which ensures diagonal dominance of $\mathbf{W}_\ell^Q \mathbf{W}_\ell^{K\top}$), $\mathbf{K}_\ell$ is well-conditioned.*

The detailed proof is provided in Section A.4.3. The intuition behind this proposition is that if the largest singular value of perturbation term $\mathbf{E}_\ell$ is smaller than the smallest singular value of the dominant term $\mathbf{B}_\ell$, then $\kappa(\mathbf{K}_\ell) \approx \kappa(\mathbf{B}_\ell)$.

> **Takeaway**
> By initializing $\mathbf{W}_\ell^V \mathbf{W}_\ell^O$ to be scaled orthonormal and $\mathbf{W}_\ell^Q \mathbf{W}_\ell^{K\top}$ to be a diagonally dominant structure, we improve the conditioning of the network Jacobian, addressing the main barrier that has historically prevented the training of completely skipless transformers.

# 6 EXPERIMENTS

We evaluate our methods in supervised learning and self-supervised learning settings. All of our experiments will be on Vision Transformers (ViTs) (Dosovitskiy et al., 2020), which have emerged as powerful models in the field of computer vision, demonstrating remarkable performance across various tasks.

## 6.1 SUPERVISED LEARNING WITH SKIPLESS VIT

We first validate our skipless ViTs on supervised learning image classification tasks. The model in this subsection is ViT-Base (12 layers, 12 heads, head dimension 64, token dimension 768). The skip models are standard ViT-Base, while in the skipless models we remove all skip connections (from both the SABs and FFNs), and use the proposed initialization for the SA weights (choosing $\alpha = 2, \beta = 0.6$ and $c = 3$)[2]. The scaled-corrected uniform orthogonal initialization (Martens et al., 2021) is used for the MLP parameters. Our implementation follows the setup in (Xu et al., 2024), except that for a fair comparison we disable the drop path, which is not applicable in skipelss models. All experiments are conducted on the ImageNet-1k (Russakovsky et al., 2015) dataset. We further compare the performance when using AdamW (Loshchilov & Hutter, 2019) and SOAP (Vyas et al., 2025) optimizers.

---

[2]We observed that our initialization hyperparameters $(\alpha, \beta, c)$ are not highly sensitive.

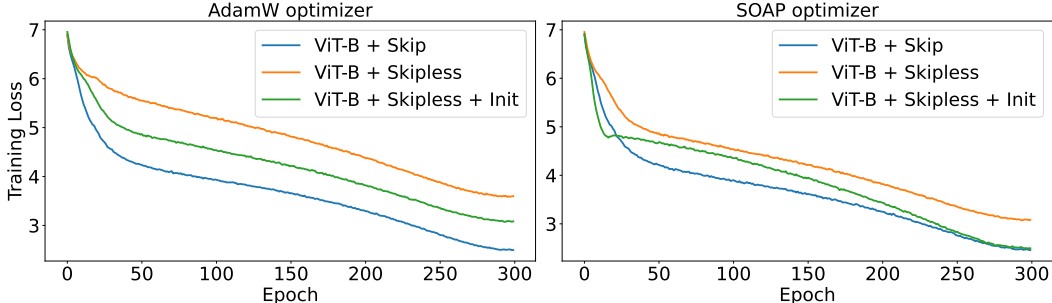

Figure 1: Supervised training loss of ViT-Base using AdamW (**Left**) and SOAP (**Right**) optimizers.

Table 1: Validation accuracy of ViT-Base on ImageNet-1k using AdamW and SOAP optimizers.

|  | Model | Accuracy |
|---|---|---|
| skip | ViT-Base + AdamW | 80.3% |
|  | ViT-Base + SOAP | 80.1% |
| skipless | ViT-Base + AdamW | 61.4% |
|  | ViT-Base + SOAP | 77.0% |
| skipless + our init | ViT-Base + AdamW | 78.1% |
|  | ViT-Base + SOAP | 80.8% |

**Results and Analysis.** As shown in Table 1, the removal of skip connections severally hampers the convergence of ViT-Base when trained with AdamW. This is evident both in the substantial accuracy drop (61.4% *vs*. 80.3%), and the high training loss with slow convergence illustrated in Fig. 1 (left). Using SOAP can partially alleviate this issue, enabling skipless models to converge more reliably and recover much of the lost performance, while they still underperform standard ViTs with skip connections. Incorporating our proposed initialization significantly mitigates these issues. When trained with AdamW, skipless ViT-Base recovers most of the lost performance. Moreover, when combined with SOAP, skipless models can converge as fast as vanilla ViT-based at the standard 300 epochs and achieve 80.8% accuracy, surpassing the skip-based ViT-Base baseline by 0.5%. These results demonstrate that the proposed initialization is essential for enabling competitive training of skipless ViTs across optimizers.

## 6.2 SELF-SUPERVISED LEARNING WITH SKIPLESS VIT

We further evaluate our skipless ViT model in the self-supervised setting. Specifically, we adopt DINO (Caron et al., 2021), a widely used self-supervised framework based on self-distillation without annotations. Here we use ViT-Small (12 layers, 6 heads, head dimension 64, token dimension 384), and $\alpha = 1.8, \beta = 1, c = 3$ for the initialization parameters, and otherwise follow similar model recipe to the previous subsection. We compare results under both AdamW and SOAP optimizers. For quantitative evaluation, we extract representations from individual or multiple blocks of frozen pre-trained models, and assess them on two downstream tasks: dense linear probing segmentation (in Section 6.2.1) and object discovery (in Section 6.2.2). For qualitative evaluation (in Section 6.3), we use Principle Component Analysis (PCA) (Abdi & Williams, 2010) to project the learned representations into 3-channel feature maps, visualized as RGB images.

### 6.2.1 DENSE LINEAR PROBING SEGMENTATION

We evaluate linear probing on dense features for the semantic segmentation task. A linear classifier is trained on top of the representation, with performance measured by mean intersection-over-union (mIoU) on PASCAL VOC2012 (Everingham et al., 2015), ADE20K (Zhou et al., 2019), and COCO-Stuff (Caesar et al., 2018) datasets. We sweep over learning rates and train for 30 epochs. For ADE20k and COCO-Stuff, we randomly sample 3,000 training images due to resource constraints.

**Results and Analysis.** As shown in Table 2, our skipless DINO ViT-Small models trained with the AdamW optimizer achieve higher performance than their skip-based counterparts on the VOC2012 and COCOStuff benchmarks when evaluated with the representation extracted from the single layer.

Table 2: Pretrained DINO ViT-Small models for 300 epochs. We also evaluate the checkpoint at 200 epochs for skipless models. Performance on linear probing segmentation tasks on different datasets.

| Epochs → | VOC2012 | | | COCOStuff | | | ADE20K | | |
|---|---|---|---|---|---|---|---|---|---|
| | 300 skip | 300 skipless | 200 skipless | 300 skip | 300 skipless | 200 skipless | 300 skip | 300 skipless | 200 skipless |
| *single feature* | | | | | | | | | |
| AdamW | 56.3 | **62.3** | 62.1 | 24.6 | 24.9 | **28.3** | **23.7** | 22.5 | 22.8 |
| SOAP | 51.3 | 57.6 | **63.4** | 21.3 | 23.5 | **27.6** | 20.5 | 21.3 | **22.5** |
| *multiscale* | | | | | | | | | |
| AdamW | 61.6 | **65.4** | 65.0 | 26.7 | 28.0 | **28.2** | 26.0 | 26.3 | **27.0** |
| SOAP | 61.3 | 59.5 | **64.8** | 25.9 | 25.1 | **27.6** | 25.3 | 23.7 | **25.6** |

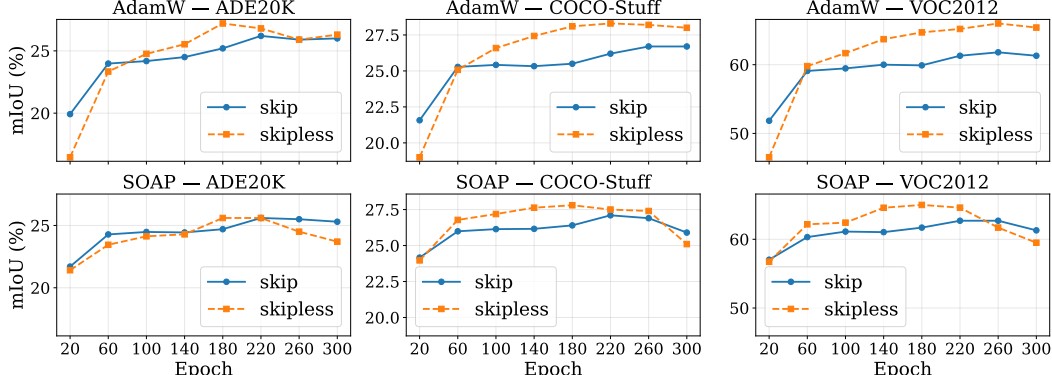

Figure 2: Performance of dense linear probing segmentation results using skip and skipless DINO ViT-Small models with AdamW and SOAP optimizers throughout the pretraining. The range of y-axis is the same for per column.

In contrast, these models show reduced accuracy on the ADE20K dataset under the same single-layer setting. We attribute this to the greater scene complexity in ADE20K, where multi-scale information is critical. The skip-based models can implicitly mix representations across layers, providing a form of multi-scale context. However, our skipless models enforce a stricter hierarchical structure, yielding more abstract features at each layer. Based on this, when multiscale layer features are explicitly aggregated at evaluation time, skipless models once again surpass their skip-connected counterparts. While training with SOAP, overall we observe the performance drops for both skip and skipless models and we conjecture that is due to the inductive bias of optimizers (Pascanu et al., 2025). Further, we demonstrate the depth analysis in Table 3. We train our models using the AdamW optimizer with different depths for 300 epochs and evaluate using the checkpoint at 200 epochs. Our models with 10 blocks perform comparably with skip models.

### 6.2.2 OBJECT DISCOVERY

Detecting salient objects is a fundamental problem in computer vision with applications in real-world vision systems. Traditional methods rely on supervised learning using large-scale high-quality annotated data, which is expensive and time-consuming to obtain these annotations (Loshchilov & Hutter, 2019). To address this challenge, recent works (Siméoni et al., 2021; Wang et al., 2023) have explored self-supervised pre-trained models, which produce high-quality and abstraction feature representations without requiring manual labels. In this subsection, we validate our pretrained DINO models using TokenCuT (Wang et al., 2023), a graph-based algorithm that leverages self-supervised transformer features for salient object detection. Following prior observations (Amir et al., 2022) that positional information gradually diminishes across layers, we compare representations from different transformer blocks and report the best-performing results. We use VOC2012 (Everingham et al., 2015) and COCO20k (Lin et al., 2014) as the evaluation datasets.

**Results and Analysis.** As shown in Fig. 3, our skipless models consistently outperform their skip-connected counterparts by a substantial margin on both the VOC2012 and COCO20K datasets under both AdamW and SOAP optimization, indicating that the representations from skipless models are

Figure 3: Pretrained DINO ViT-Small models for 300 epochs. For skipless models, we also evaluated checkpoint at 200 epochs. Performance on object discovery tasks using TokenCut on VOC2012 and COCO20k datasets.

| Epoch → Optimizer ↓ | VOC2012 | | | COCO20k | | |
| --- | --- | --- | --- | --- | --- | --- |
| | 300 skip | 300 skipless | 200 skipless | 300 skip | 300 skipless | 200 skipless |
| AdamW | 32.3 | 53.5 | **54.0** | 21.2 | 36.5 | **38.5** |
| SOAP | 49.4 | 63.2 | **68.1** | 27.5 | 46.7 | **54.1** |

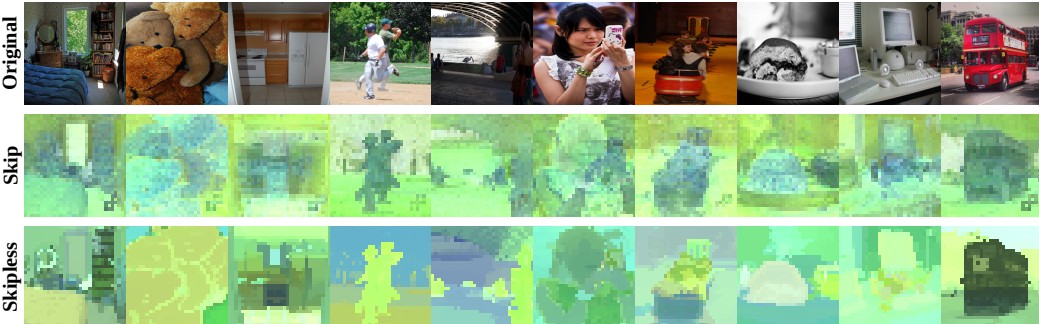

Figure 4: Visualize learned representations from pretrained DINO models **without cherry-picking**.

abstract and high-quality. Furthermore, in Table 4, we evaluate end-to-end trained models of varying depths and find that skipless ViTs with only 9 layers surpass skip-based 12 layer ViTs on both datasets, highlighting the efficiency of the skipless design.

### 6.3 QUALITATIVE EVALUATION

To deeply analyze the effectiveness of our skipless ViTs, we visualize representations of pre-trained models. Here, we choose the features from 11-th blocks. As shown in Fig. 4, we select the first 40 images from the COCO validation set without cherry-picking. Ten examples are shown in the main paper, and the rest are shown in Fig. 7. PCA is applied to project the representations into three channels and render them as RGB images. The figure clearly demonstrates that in models with skip connections, the features appear patchy and noisy, as shallow information is repeatedly injected into deeper layers, hindering the learning of high-level semantic representations. In contrast, skipless models yield clearer object boundaries between different semantic regions with more consistent colors within the same object. These results suggest that skipless ViTs capture more abstract and semantically coherent features.

## 7 DISCUSSION AND LIMITATION

Our experiments focus on Vision Transformers, as they offer a well-understood and widely used backbone for analysis, visualization, and controlled experimentation. The inherently compositional structure of vision tasks also makes this domain particularly suitable for examining how skipless models form hierarchical representations. Due to computational constraints, our evaluations are limited to ViT models in the 100M-parameter range. We expect that extending these insights to larger-scale architectures, such as billion-parameter models, will provide further understanding of skipless training at scale and represents an exciting direction for future work.

Although our conditioning analysis relies on a mild block incoherence assumption that we do not explicitly verify. The strong and consistent empirical performance of our initialization across different model depths and dense prediction tasks suggests that this theoretical simplification remains practically meaningful.

## 8 CONCLUSION

In this paper, we present a theoretical analysis of the transformer Jacobian and, building on this first principle, propose a theoretically grounded initialization scheme that requires no architectural mod-

ifications. This scheme enables efficient training of skipless Vision Transformers. Furthermore, our skipless models outperform their residual-based counterparts on dense prediction tasks, suggesting that they learn more abstract and higher-quality internal representations. We hope our work provides new insights into hierarchical representation learning in vision.

## ETHICS STATEMENT

This work uses only publicly available benchmark datasets and does not involve human subjects, personally identifiable information, or sensitive data. Our methods are intended for advancing fundamental research in machine learning.

## REPRODUCIBILITY STATEMENT

We have taken care to ensure the reproducibility of all results presented in this paper. Where external code was used, explicit references are provided. We will release the code upon publication.

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

## A APPENDIX

### A.1 USE OF LLMs

Large language models (LLMs) were used to assist with proofreading, formatting, and improving the clarity of writing. All technical contributions, experiments, and analyses were designed and conducted by the authors

### A.2 END-TO-END TRAINING WITH LESS DEPTH

In this section, we train end-to-end skipless DINO ViT-Small models for 300 epochs with varied depth using AdamW optimizer and evaluate on linear probing semantic segmentation and object discovery tasks. We use the checkpoint at 200 epochs.

Table 3: End-to-end training performance on dense linear probing segmentation on our models with varied depth (AdamW).

|          | depth | VOC2012 | COCOStuff | ADE20K |
|----------|-------|---------|-----------|--------|
| skip     | 12    | 61.6    | 26.7      | 26.0   |
| skipless | 12    | 65.0    | 28.2      | 27.0   |
|          | 11    | 66.2    | 28.0      | 26.7   |
|          | 10    | 64.1    | 27.1      | 25.2   |
|          | 9     | 61.1    | 26.0      | 24.4   |

Table 4: End-to-end training performance on object discovery on our models with varied depth (AdamW).

|          | depth | VOC2012 | COCO20k |
|----------|-------|---------|---------|
| skip     | 12    | 32.3    | 21.2    |
| skipless | 12    | 53.5    | 36.5    |
|          | 11    | 47.4    | 31.9    |
|          | 10    | 43.9    | 25.4    |
|          | 9     | 34.8    | 24.0    |

### A.3 JACOBIAN

In this section, we provide the derivation of $\mathbf{K}_\ell$.

The derivative of the SA output with respect to the input is

$$
\begin{aligned}
\frac{\partial \text{vec}\left(\text{SA}_\ell(\hat{\mathbf{X}}_{\ell-1})\right)}{\partial \hat{\mathbf{x}}^{(\ell-1)}} &= \frac{\partial \text{vec}\left(\mathbf{A}_\ell(\hat{\mathbf{X}}_{\ell-1})\mathbf{V}_\ell\right)}{\partial \hat{\mathbf{x}}^{(\ell-1)}} \\
&= (\mathbf{V}_\ell^\top \otimes \mathbf{I}_n)\mathbf{A}_\ell' + (\mathbf{I}_d \otimes \mathbf{A}_\ell)\frac{\partial \text{vec}\left(\mathbf{V}_\ell\right)}{\partial \hat{\mathbf{x}}^{(\ell-1)}} \\
&= (\mathbf{V}_\ell^\top \otimes \mathbf{I}_n)\mathbf{A}_\ell' + (\mathbf{I}_d \otimes \mathbf{A}_\ell)(\mathbf{W}_\ell^{V^\top} \otimes \mathbf{I}_{n\times n}) \\
&= (\mathbf{V}_\ell^\top \otimes \mathbf{I}_n)\mathbf{A}_\ell' + \mathbf{W}_\ell^{V^\top} \otimes \mathbf{A}_\ell
\end{aligned}
\tag{12}
$$

The Jacobian of the attention matrix to the input is

$$\begin{aligned}
\mathbf{K}_l &= \frac{\partial f_{\text{SA}}(\hat{\mathbf{x}}^{(\ell-1)}; \theta^{(\ell)})}{\partial \hat{\mathbf{x}}^{(\ell-1)}} \in \mathbb{R}^{nd \times nd}, \\
&= \frac{\partial \text{vec}\left(\text{SA}_\ell\left(\mathbf{X}_{\ell-1}\right) \mathbf{W}_\ell^{\text{O}}\right)}{\partial \hat{\mathbf{x}}^{(\ell-1)}} \\
&= \frac{\partial \text{vec}\left(\text{SA}_\ell(\mathbf{X}_{\ell-1}) \mathbf{W}_\ell^{\text{O}}\right)}{\partial \text{vec}\left(\text{SA}_\ell(\mathbf{X}_{\ell-1})\right)} \frac{\partial \text{vec}\left(\text{SA}_\ell(\mathbf{X}_{\ell-1})\right)}{\partial \hat{\mathbf{x}}^{(\ell-1)}} \\
&= (\mathbf{W}_\ell^{\text{O}\top} \otimes \mathbf{I}_n) \frac{\partial \text{vec}\left(\text{SA}_\ell(\mathbf{X}_{\ell-1})\right)}{\partial \hat{\mathbf{x}}^{(\ell-1)}} \\
&= (\mathbf{W}_\ell^{\text{O}\top} \otimes \mathbf{I}_n) \left((\mathbf{V}_\ell^\top \otimes \mathbf{I}_n)\mathbf{A}_\ell' + \mathbf{W}_\ell^{\text{V}\top} \otimes \mathbf{A}_\ell\right) \\
&= ((\mathbf{W}_\ell^{\text{O}\top}\mathbf{V}_\ell^\top \otimes \mathbf{I}_n)\mathbf{A}_\ell' + \mathbf{W}_\ell^{\text{O}\top}\mathbf{W}_\ell^{\text{V}\top} \otimes \mathbf{A}_l) \\
&= (\hat{\mathbf{X}}_{\ell-1}\mathbf{W}_\ell^{\text{V}}\mathbf{W}_\ell^{\text{O}} \otimes \mathbf{I}_n)^\top \mathbf{A}_\ell' + (\mathbf{W}_\ell^{\text{V}}\mathbf{W}_\ell^{\text{O}})^\top \otimes \mathbf{A}_\ell
\end{aligned} \tag{13}$$

## A.4 PROOF

### A.4.1 SOFTMAX CONDITIONING

In this section, we provide the empirical demonstration of the Proposition 1. We conduct a simple simulation experiment and the result is shown in Fig. 5. We choose a square matrix $\mathbf{M} \in \mathbb{R}^{10 \times 10}$ and set $\alpha = 0.1, \beta = 5$ for the "peak" case and $\alpha = 0.1, \beta = 0$ for the "diffuse" case. Empirically, we can see that when choosing large $\beta$ (ensuring diagonal dominance), the softmax produces a near identity matrix with $\kappa \approx 1.1$. However, if $\mathbf{M}$ is truncated normal initialized, each row of the softmax output is near uniform and the output is ill-conditioned with $\kappa \approx 730.1$.

Distribution of $\mathbf{X}\mathbf{X}^\top$ and $\mathbf{X}\mathbf{Z}\mathbf{X}^\top$

Given $\mathbf{X} \in \mathbb{R}^{n \times d} \sim \mathcal{N}(0, \mathbf{I})$, we have the mean and variance of $\mathbf{A} = \mathbf{X}\mathbf{X}^\top$ as follows,

- Diagonal entries $(i = j)$:

$$\mathbf{A}_{ii} \sim \chi_d^2, \qquad \mathbb{E}[\mathbf{A}_{ii}] = d, \qquad \text{Var}(\mathbf{A}_{ii}) = 2d, \tag{14}$$

where $\chi$ is Wishart distribution.

- Off-diagonal entries $(i \neq j)$:

$$\mathbb{E}[\mathbf{A}_{ij}] = 0, \qquad \text{Var}(\mathbf{A}_{ij}) = d, \tag{15}$$

Then given $\mathbf{Z} \sim \mathcal{N}(0, \frac{1}{d}\mathbf{I})$, we have the mean and variance of $\mathbf{B} = \mathbf{X}\mathbf{Z}\mathbf{X}^\top$ as follows,

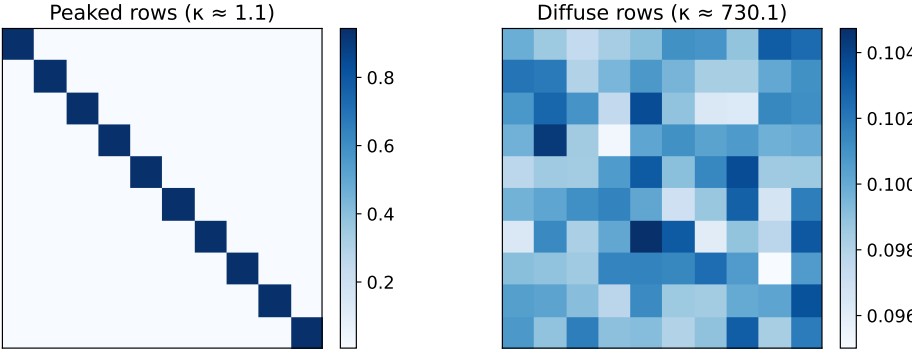

Figure 5: **Left:** We choose $\alpha = 0.1, \beta = 5$ to ensure diagonal dominance. **Right:** We choose $\alpha = 0.1, \beta = 0$.

- Diagonal entries ($i = j$):

$$\mathbb{E}[\mathbf{B}_{ii}] = 0, \qquad \text{Var}(\mathbf{B}_{ii}) \approx d + 2. \tag{16}$$

- Off-diagonal entries ($i \neq j$):

$$\mathbb{E}[\mathbf{B}_{ij}] = 0, \qquad \text{Var}(\mathbf{B}_{ij}) \approx 1. \tag{17}$$

For the combined matrix (attention map) $\mathbf{C} = \alpha\mathbf{B} + \beta\mathbf{A}$, we have

- Diagonal entries ($i = j$):

$$\mathbb{E}[\mathbf{C}_{ii}] = \beta d, \qquad \text{Var}(\mathbf{C}_{ii}) \approx \alpha^2(d+2) + \beta^2(2d). \tag{18}$$

- Off-diagonal entries ($i \neq j$):

$$\mathbb{E}[\mathbf{C}_{ij}] = 0, \qquad \text{Var}(\mathbf{C}_{ij}) \approx \alpha^2 + \beta^2 d. \tag{19}$$

The default initialization is equivalent to $\beta = 0$ (no diagonal dominance in weight initialization) and $\alpha = \mathcal{O}(\frac{1}{d})$ (usually around 0.04). All the values in the attention map are mean 0 with a variance much smaller than 1, which satisfy the diffuse condition.

When $\beta > 0$, the difference between the diagonal elements and an off-diagonal element $\gamma = \mathbf{C}_{ii} - \mathbf{C}{ij}$ follows

$$\gamma \sim \mathcal{N}(\beta d, \alpha^2(d+3) + \beta^2(3d)) \tag{20}$$

(The covariance between $\mathbf{C}_{ii}$ and $\mathbf{C}{ij}$ is close to 0 when $d$ is large.), which satisfy the diagonal dominent condition with a proper $\beta$.

To justify our assumption that the token embeddings $\mathbf{X} \in \mathbb{R}^{n \times d} \sim \mathcal{N}(0, \mathbf{I})$, we show that their empirical distribution closely matches a zero-mean, approximately isotropic Gaussian (shown in Fig. 6).

### A.4.2 PROOF OF PROPOSED INITIALIZATION

**Lemma 1.** *(Jacobian of the softmax function) Let $\mathbf{A} = \text{softmax}(\mathbf{M})$, where $\mathbf{M} = \mathbf{X}\mathbf{W}^Q\mathbf{W}^{K^\top}\mathbf{X}^\top$. With the proposed initialization (see in Eq. (11)), we can show that the 2-norm of the derivative scales, $\left\|\frac{\partial \text{vec}(\mathbf{A})}{\partial \text{vec}(\mathbf{X})}\right\|_2 = O\left(\alpha e^{-\beta}\right)$*

*Proof.* See the derivation in Eq. (12), we have the bound:

$$\left\|\frac{\partial \text{vec}(\mathbf{A})}{\partial \text{vec}(\mathbf{X})}\right\|_2 \leq \left\|\frac{\partial \text{vec}(\mathbf{A})}{\partial \text{vec}(\mathbf{M})}\right\|_2 \tag{21}$$

Since $\text{softmax}$ is a row normalization, the Jacobian $\mathbf{J}_{\mathbf{A}} = \frac{\partial \text{vec}(\mathbf{A})}{\partial \text{vec}(\mathbf{M})} \in \mathbb{R}^{n^2 \times n^2}$ is a block diagonal matrix. For each block $i$, we have:

$$(\mathbf{J}_{\mathbf{A},i})_{jk} = \mathbf{A}_{ij}(\delta_{jk} - \mathbf{A}_{ik}), \tag{22}$$

where $\mathbf{A}_{ij}$ is the $i$-th row and $j$-th column entry of $\mathbf{A}$.

Obviously, using our proposed initialization (laerger $\beta$ leads to more diagonally dominant), we have:

$$\mathbf{A}_{ii} = 1 - \mathcal{O}(\alpha e^{-\beta}) \quad \text{and} \quad \mathbf{A}_{ij} = \mathcal{O}(\alpha e^{-\beta}) \tag{23}$$

Based on this, we analyze the order of magnitude of the elements in $(\mathbf{J}_{\mathbf{A},i})$. In the case of $j = k$:

$$(\mathbf{J}_{\mathbf{A},i})_{ii} = \mathbf{A}_{ii}(1 - \mathbf{A}_{ii}) = \mathcal{O}(\alpha e^{-\beta}), \text{if } i = j \tag{24}$$

$$(\mathbf{J}_{\mathbf{A},i})_{ii} = \mathbf{A}_{ii}(1 - \mathbf{A}_{ii}) = \mathcal{O}(\alpha e^{-\beta}), \text{if } i \neq j \tag{25}$$

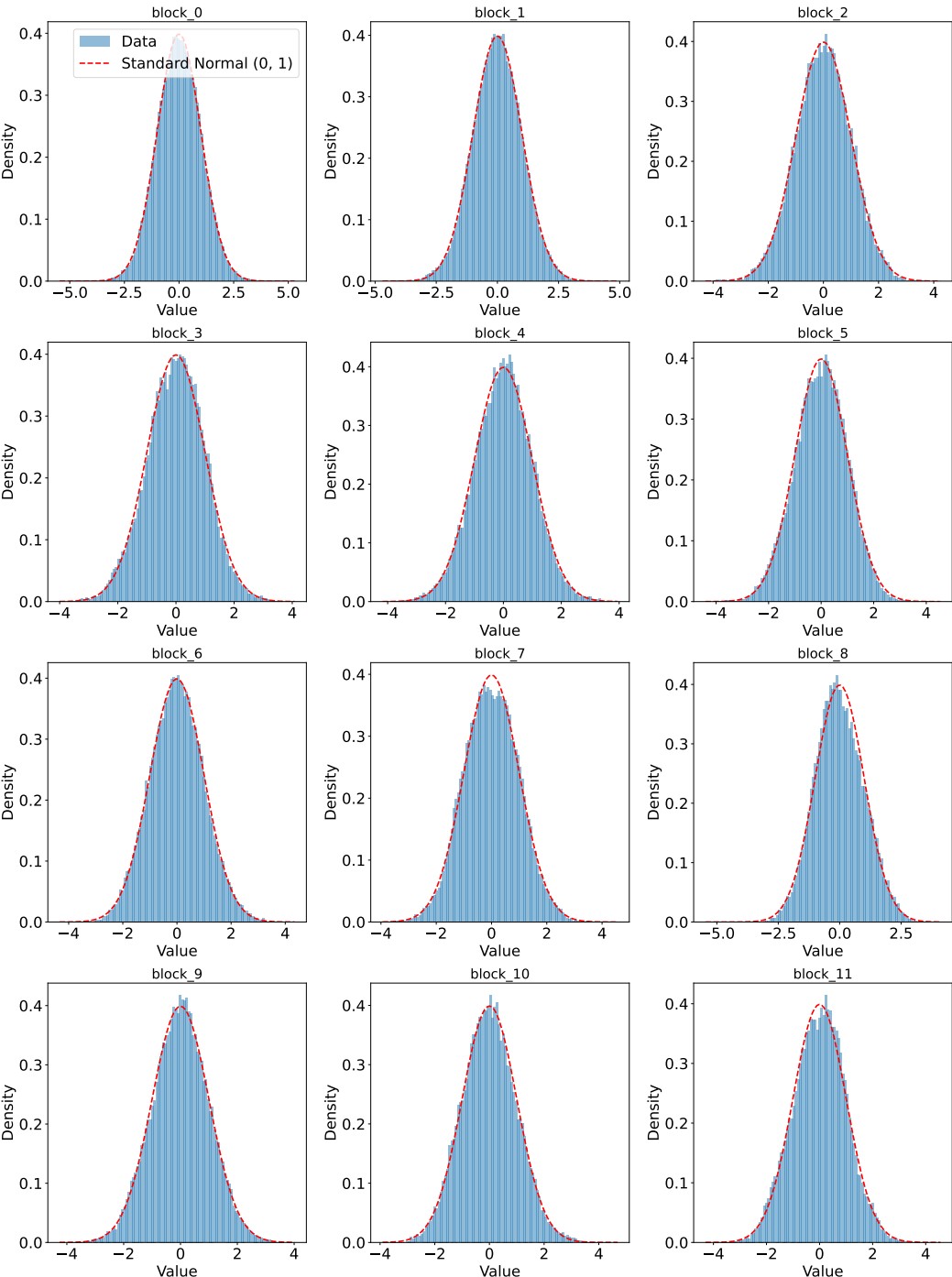

Figure 6: Distribution of token embeddings after the pre-layer norm throughout the blocks using proposed initialization.

In the case of $j \neq k$:

$$(\mathbf{J}_{\mathbf{A},i})_{jk} = -\mathbf{A}_{ij}\mathbf{A}ik = \mathcal{O}(\alpha e^{-\beta}) \tag{26}$$

Then we have the bound for $\|\mathbf{J}_{\mathbf{A},i}\|$:

$$\|\mathbf{J}_{\mathbf{A},i}\|_F = \sqrt{\sum_{j,k=1}^{d} ((\mathbf{J}_{\mathbf{A},i})_{jk})^2} = d \cdot \mathcal{O}(\alpha e^{-\beta}) \tag{27}$$

Therefore, the 2-norm is:

$$\|\mathbf{J}_{\mathbf{A},i}\|_2 \leq \|\mathbf{J}_{\mathbf{A},i}\|_F = \mathcal{O}(\alpha e^{-\beta}) \tag{28}$$

Then we have :

$$\left\| \frac{\partial \mathrm{vec}(\mathbf{A})}{\partial \mathrm{vec}(\mathbf{X})} \right\|_2 = \max_i \|\mathbf{J}_{\mathbf{A},i}\|_2 = \mathcal{O}(\alpha e^{-\beta}) \tag{29}$$

$\square$

### A.4.3 CONDITIONING OF $\mathbf{K}_\ell$

In this section we provide the proof for Proposition 2

*Proof.* We show that, with proposed initialization, $\mathbf{A}$ is well conditioned such that $\kappa(\mathbf{A}) \approx 1$

Next step, we show that the term $\mathbf{E}$ is a small perturbation of Jacobian $\mathbf{J}$.

Bound $\|\mathbf{E}\|_2$ using norm submultiplicativity:

$$\|\mathbf{E}\|_2 \leq \|\mathbf{W}_\ell^O\|_2 \|\mathbf{W}_\ell^V\|_2 \|\mathbf{X}_{\ell-1}\|_2 \left\| \frac{\partial(\mathrm{vec}(\mathbf{A}_\ell(\mathbf{X}_{\ell-1})))}{\partial \mathbf{x}^{\ell-1}} \right\|_2 \tag{30}$$

Since $\|\mathbf{W}_\ell^O \mathbf{W}_\ell^V\|_2 \leq \|\mathbf{W}_\ell^O\|_2 \|\mathbf{W}_\ell^V\|_2 = 1$ and $\|\mathbf{X}\|_2$ is bounded, using Lemma 1, we have:

$$\|\mathbf{W}_\ell^O \mathbf{W}_\ell^V\|_2 \leq \mathcal{O}(\alpha e^{-\beta}) \tag{31}$$

Combine the bound $\|\mathbf{B}\|_2 \approx 1$ and $\|\mathbf{E}\|_2 \leq \mathcal{O}(\alpha e^{-\beta})$, and there exists $\alpha$ and $\beta$ such that $\|\mathbf{E}\|_2 \ll \|\mathbf{B}\|_2$. Therefore $\mathbf{E}$ is a small perturbation of $\mathbf{B}$ and $\mathbf{J}$ is well-conditioned $\kappa(\mathbf{J}) \approx 1$.

$\square$

### A.4.4 CONDITION OF MATRIX CONCATENATION

$$\text{Let } \mathbf{M} = [\mathbf{A} \ \mathbf{B}] \in \mathbb{R}^{n \times (d_1 + d_2)}.$$

Denote the spectral norms and minimal singular values by

$$s_{\max} = \max\{\sigma_{\max}(\mathbf{A}), \sigma_{\max}(\mathbf{B})\}, \quad s_{\min} = \min\{\sigma_{\min}(\mathbf{A}), \sigma_{\min}(\mathbf{B})\},$$

and the mutual coherence parameter measuring alignment between $\mathbf{A}$ and $\mathbf{B}$ and a balanced condition $\tau$ measuring the norm difference of the matrices:

$$\rho := \|\mathbf{A}^\top \mathbf{B}\|_2, \tag{32}$$

$$\tau := \frac{\max\{\|\mathbf{A}\|_2, \|\mathbf{B}\|_2\}}{\min\{\|\mathbf{A}\|_2, \|\mathbf{B}\|_2\}} = \frac{\max\{\sigma_{\max}(\mathbf{A}), \sigma_{\max}(\mathbf{B})\}}{\min\{\sigma_{\max}(\mathbf{A}), \sigma_{\max}(\mathbf{B})\}}. \tag{33}$$

For any nonzero vectors $\mathbf{x} \in \mathbb{R}^{d_1}, \mathbf{y} \in \mathbb{R}^{d_2}$ we have

$$\frac{\|\mathbf{M} \begin{bmatrix} \mathbf{x} \\ \mathbf{y} \end{bmatrix}\|^2}{\|\begin{bmatrix} \mathbf{x} \\ \mathbf{y} \end{bmatrix}\|^2} = \frac{\begin{bmatrix} \mathbf{x} \\ \mathbf{y} \end{bmatrix}^\top \mathbf{M}^\top \mathbf{M} \begin{bmatrix} \mathbf{x} \\ \mathbf{y} \end{bmatrix}}{\|\mathbf{x}\|^2 + \|\mathbf{y}\|^2} = \frac{\|\mathbf{A}\mathbf{x}\|^2 + \|\mathbf{B}\mathbf{y}\|^2 + 2\langle \mathbf{A}\mathbf{x}, \mathbf{B}\mathbf{y} \rangle}{\|\mathbf{x}\|^2 + \|\mathbf{y}\|^2}. \tag{34}$$

Hence the largest singular value of $\mathbf{M}$ is

$$\begin{aligned} \sigma_{\max}(\mathbf{M})^2 &= \max_{\mathbf{x},\mathbf{y}\neq 0} \frac{\|\mathbf{A}\mathbf{x}\|^2 + \|\mathbf{B}\mathbf{y}\|^2 + 2\langle \mathbf{A}\mathbf{x}, \mathbf{B}\mathbf{y} \rangle}{\|\mathbf{x}\|^2 + \|\mathbf{y}\|^2} \\ &\leq \max_{\mathbf{x},\mathbf{y}\neq 0} \frac{\sigma_{\max}(\mathbf{A})^2\|\mathbf{x}\|^2 + \sigma_{\max}(\mathbf{B})^2\|\mathbf{y}\|^2 + 2\rho\|\mathbf{x}\|\|\mathbf{y}\|}{\|\mathbf{x}\|^2 + \|\mathbf{y}\|^2} \\ &\leq \max\{\sigma_{\max}(\mathbf{A})^2, \sigma_{\max}(\mathbf{B})^2\} + \max_{\mathbf{x},\mathbf{y}\neq 0} \frac{2\rho\|\mathbf{x}\|\|\mathbf{y}\|}{\|\mathbf{x}\|^2 + \|\mathbf{y}\|^2} \\ &\leq s_{\max}^2 + \rho. \end{aligned} \tag{35}$$

The smallest singular value of $\mathbf{M}$ is

$$\begin{aligned} \sigma_{\min}(\mathbf{M})^2 &= \min_{\mathbf{x},\mathbf{y}\neq 0} \frac{\|\mathbf{A}\mathbf{x}\|^2 + \|\mathbf{B}\mathbf{y}\|^2 + 2\langle \mathbf{A}\mathbf{x}, \mathbf{B}\mathbf{y} \rangle}{\|\mathbf{x}\|^2 + \|\mathbf{y}\|^2} \\ &\geq \min_{\mathbf{x},\mathbf{y}\neq 0} \frac{\sigma_{\min}(\mathbf{A})^2\|\mathbf{x}\|^2 + \sigma_{\min}(\mathbf{B})^2\|\mathbf{y}\|^2 - 2\rho\|\mathbf{x}\|\|\mathbf{y}\|}{\|\mathbf{x}\|^2 + \|\mathbf{y}\|^2} \\ &\geq \min\{\sigma_{\min}(\mathbf{A})^2, \sigma_{\min}(\mathbf{B})^2\} + \min_{\mathbf{x},\mathbf{y}\neq 0} \frac{-2\rho\|\mathbf{x}\|\|\mathbf{y}\|}{\|\mathbf{x}\|^2 + \|\mathbf{y}\|^2} \\ &\geq s_{\min}^2 - \rho. \end{aligned} \tag{36}$$

Therefore

$$\kappa(\mathbf{M}) = \frac{\sigma_{\max}(\mathbf{M})}{\sigma_{\min}(\mathbf{M})} \leq \sqrt{\frac{s_{\max}^2 + \rho}{s_{\min}^2 - \rho}} \leq \sqrt{\frac{1 + \frac{\rho}{s_{\max}^2}}{1 - \frac{\rho}{s_{\min}^2}}} \cdot \frac{s_{\max}}{s_{\min}} \leq \tau \sqrt{\frac{1 + \frac{\rho}{s_{\max}^2}}{1 - \frac{\rho}{s_{\min}^2}}} \, \kappa_{\max}, \tag{37}$$

where $\kappa_{\max}$ is the largest condition number of the component matrices.

Under a mild block-incoherence condition (i.e., $\rho \to 0$), and balanced blocks ($\tau \to 1$), the concatenated condition number is controlled by the worst block condition number $\kappa_{\max}$.

### A.5 VISUALIZATION

As shown in Fig. 7, the skipless DINO ViT-Small produces representations that form much more semantically coherent clusters. Moreover, Fig. 8 reveals a clear hierarchical progression: earlier layers capture meaningful subparts and mid-level structures, while deeper layers focus on complete objects. Such hierarchical patterns are weaker and diffuse in the residual baselines.

### A.6 DINO PRETRAINING LOSS

In Fig. 9, we demonstrate the DINO pretraining loss for both skip DINO ViT-Small and skipless DINO ViT-Small with proposed method.

### A.7 JACOBIAN CONDITIONING DURING TRAINING

We tracked the condition numbers of the full Jacobian ($\kappa(\mathbf{J}\mathbf{J}^\top)$) as well as the per-layer Jacobian kernels during DINO ViT-Small training. As shown in Table 5, the standard skipless baseline diverges early in training, resulting in numerical overflow. In contrast, our *skipless + init* variant maintains a condition number of the same order of magnitude as the residual (skip) baseline throughout

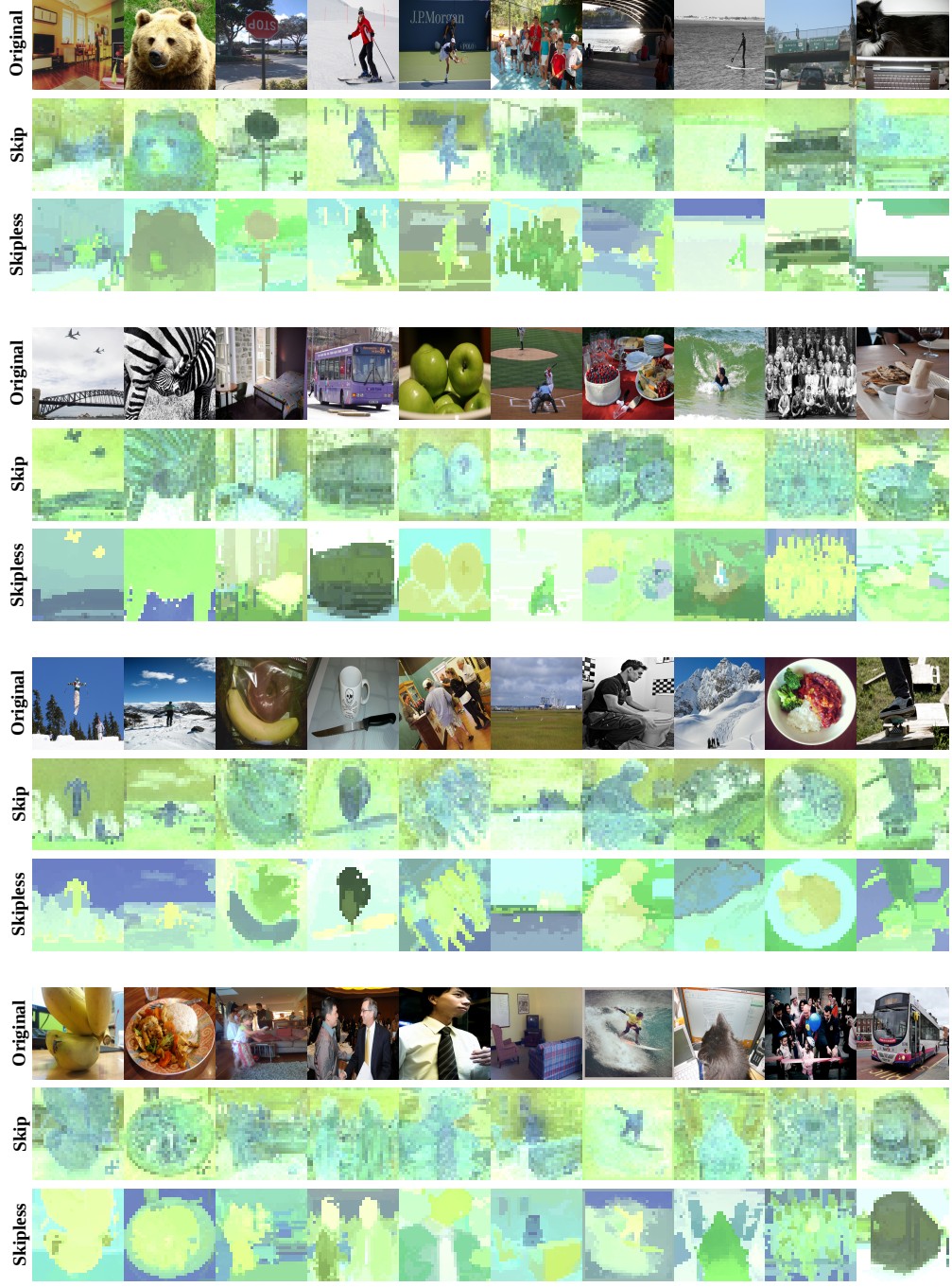

Figure 7: PCA visualization of representation

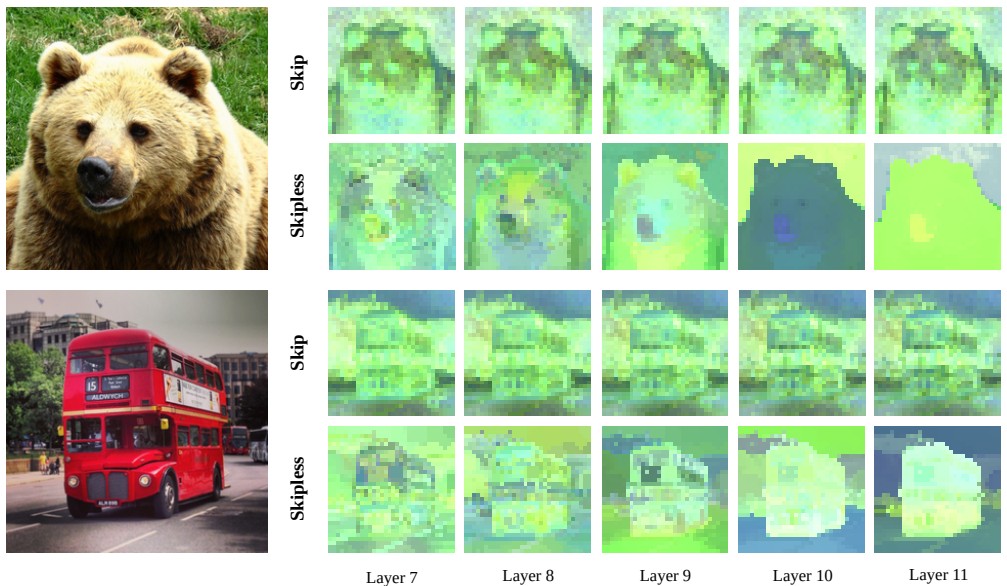

Figure 8: PCA visualization of representation from multiple layers

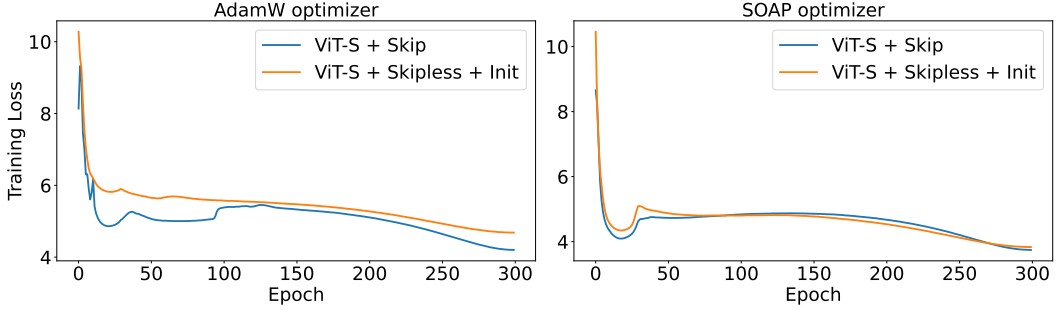

Figure 9: DINO ViT-Small Pretraining Loss

Table 5: Condition number of the full Jacobian $\kappa(\mathbf{J}\mathbf{J}^\top)$ during DINO ViT-S training.

|  | Epoch 20 | Epoch 60 | Epoch 120 | Epoch 180 | Epoch 240 | Epoch 300 |
|---|---|---|---|---|---|---|
| skip | 127 | 72 | 82 | 93 | 113 | 144 |
| skipless | $\infty$ | $\infty$ | $\infty$ | $\infty$ | $\infty$ | $\infty$ |
| skipless + init | 195 | 340 | 261 | 160 | 176 | 207 |

the entire optimization trajectory. This demonstrates that our initialization successfully mitigates the optimization pathologies that typically arise in skipless architectures.

We also analyzed the conditioning of the layer-wise Jacobian kernels, computing $\kappa(\mathbf{K}_\ell \mathbf{K}_\ell^\top)$ for skipless models and $\kappa((\mathbf{K}_\ell + \mathbf{I})(\mathbf{K}_\ell + \mathbf{I})^\top)$ for residual models. Table 6 shows results across depth and training epochs.

Table 6: Layer-wise Jacobian condition numbers across layers. We report $\kappa(\mathbf{K}_\ell \mathbf{K}_\ell^\top)$ for skipless, and $\kappa((\mathbf{K}_\ell + \mathbf{I})(\mathbf{K}_\ell + \mathbf{I})^\top)$ for skip.

| **Epoch 20** | | | | |
|---|---|---|---|---|
|  | L0 | L4 | L7 | L11 |
| skip | 1.05 | 1.49 | 1.35 | 1.04 |
| skipless | $\infty$ | $\infty$ | $\infty$ | $\infty$ |
| skipless + init | 10 | 2.7 | 7.5 | 12.5 |

| **Epoch 100** | | | | |
|---|---|---|---|---|
|  | L0 | L4 | L7 | L11 |
| skip | 1.06 | 1.54 | 1.25 | 1.25 |
| skipless | $\infty$ | $\infty$ | $\infty$ | $\infty$ |
| skipless + init | 6.25 | 3.7 | 8.54 | 19.2 |

| **Epoch 200** | | | | |
|---|---|---|---|---|
|  | L0 | L4 | L7 | L11 |
| skip | 1.03 | 1.57 | 1.21 | 1.37 |
| skipless | $\infty$ | $\infty$ | $\infty$ | $\infty$ |
| skipless + init | 6.25 | 3.84 | 11.1 | 21.7 |

| **Epoch 300** | | | | |
|---|---|---|---|---|
|  | L0 | L4 | L7 | L11 |
| skip | 1.02 | 1.38 | 1.28 | 1.51 |
| skipless | $\infty$ | $\infty$ | $\infty$ | $\infty$ |
| skipless + init | 3.44 | 3.57 | 14.3 | 20.2 |

## A.8 LANGUAGE MODELING

We extended our evaluation to Language Transformers. We pretrained a 110M parameter model on the C4 dataset for 20k steps (using AdamW) and evaluated zero-shot performance on five common-sense reasoning tasks following (He et al., 2023). We use a batch size of 256k tokens and ablate learning rate of $\{1, 3, 5, 7\} \times e^{-n}$. We use $\alpha = 0, \beta = 0.5$ for our initialization. The zero-shot performance on 5 downstream tasks are shown in Table 7. We use the `flash-linear-attention` codebase (Yang & Zhang, 2024).

Table 7: Zero-shot performance on downstream tasks.

| Architecture | BoolQ | HellaSwag | Winogrande | PIQA | SIQA | Average |
|---|---|---|---|---|---|---|
| skip | 57.3 | 29.7 | 50.9 | 63.9 | 36.5 | 47.7 |
| skipless | - | - | - | - | - | - |
| skipless + $\alpha$=0, $\beta$=0.5 | 61.2 | 28.2 | 52.2 | 62.7 | 36.5 | 48.2 |

Without our initialization, the standard skipless Transformer diverges immediately. With our method, it achieves comparable performance with the residual baseline. Crucially, while prior work (He et al., 2023) can train skipless Transformers, it requires 5 times more training steps to match residual performance. In contrast, our method achieves this at the same training speed ($1\times$ steps) as the baseline, demonstrating significantly superior efficiency.

