# OpenReview forum: "Cutting the Skip: Training Residual-Free Transformers"
_ICLR.cc/2026/Conference — ICLR 2026 Poster_

### Official Review · Reviewer_rJvD · 2025-10-28

**Soundness:** 3
**Presentation:** 3
**Contribution:** 2
**Rating:** 6
**Confidence:** 4

**Summary:**

This paper proposes an initialization to train a transformer model without skip connection. The initialization strategy is developed based on a theoretical analysis of the transformer Jacobian, which reveals that skip connections stabilize optimization by improving Jacobian conditioning. To replicate this stabilization effect without skip connections, the paper initializes the product of the projection matrices to ensure the good conditioning of attention matrix.  Results on ViT show that, when combined with SOAP optimizer, skipless models achieve comparable convergence speed and accuracy to residual-based models in supervised image classification task, and outperform residual-based models in self-supervised dense prediction tasks.

**Strengths:**

- The overall paper is well-written and easy to follow.
- The proposed initialization scheme requires no architectural modifications.
- The experimental validation is comprehensive, covering both supervised and self-supervised learning settings on ViTs. And the result aligns with the analysis and explanation.

**Weaknesses:**

- How to demonstrate "learning richer, more semantically coherent internal representations" is unclear. Additionally, the significance of training skipless transformers remains ambiguous.
- Regarding experiments across different scales, I am curious whether this method is only effective for Vision Transformers (ViTs) or applicable to transformer models with general architectures. This is because the analysis in the paper has little relevance to vision-specific characteristics.
- The paper only explores which type of initialization can lead to better conditioning of the network Jacobian, but it does not essentially guarantee better model performance.

Overall, this work represents a valuable initial attempt. However, for different transformer architectures, including those with various subsequently proposed activation functions, normalization layers, and MoE structures, the applicability and generalizability of this initialization method still require further experimental validation.

**Questions:**

- Why does an abrupt inflection point appear in Figure 1? The loss is obviously not smooth.

---

> ### Author Response · Authors · 2025-11-21
> **Response to Reviewer rJvD**
>
> Dear Reviewer rJvD,
>
> We sincerely appreciate your insightful and constructive feedback, which has been instrumental in improving our work. Please find our detailed point-by-point responses below, along with the corresponding revisions in the manuscript (highlighted in blue).
>
> > **W1: How to demonstrate "learning richer, more semantically coherent internal representations" is unclear. Additionally, the significance of training skipless transformers remains ambiguous.**
>
> To illustrate hierarchical representations, we focus on images containing a clearly identifiable dominant object. In such cases, skipless ViTs exhibit a clean progression: early layers highlight meaningful substructures, while deeper layers capture the full object structure. In contrast, these hierarchical patterns are noticeably weaker or blurrier in residual baselines. The corresponding visualizations are provided in Appendix A.5 and clearly show that the baseline ViTs admitting skip connections exhibit weaker and blurrier structures.
>
> Regarding the significance of training skipless Transformers, our motivation is to restore the model’s effective depth by removing skip connections. Prior work has shown that residual pathways substantially shorten effective depth [1,2], meaning that deeper layers may not be fully utilized. By enabling stable training without skip connections, our method forces the model to leverage its entire depth.
>
> This insight yields practical benefits. Qualitatively, skipless models learn more semantically coherent and well-clustered representations. Empirically, they perform better on dense prediction tasks such as segmentation and object discovery, where strong semantic structure is crucial. In comparison, residual models can “leak” shallow representations through skip paths, leading to less coherent feature hierarchies.
>
> - [1] Veit et al., Residual Networks Behave Like Ensembles of Relatively Shallow Networks, NIPS 2016.
>
> - [2] Csordás et al., Do Language Models Use Their Depth Efficiently? NeurIPS 2025.
>
> > **W2: Regarding experiments across different scales, I am curious whether this method is only effective for Vision Transformers (ViTs) or applicable to transformer models with general architectures. This is because the analysis in the paper has little relevance to vision-specific characteristics.**
>
> **Response:** To further demonstrate that our method is **not limited to vision**, we extended our evaluation to Language Transformers. We pretrained a ~110M parameter model on the C4 dataset for 20k steps (using AdamW) and evaluated zero-shot performance on five common-sense reasoning tasks following [1].
>
> | Architecture                 | BoolQ | HellaSwag | Winogrande | PIQA | SIQA | Average |
> |-----------------------------|:-----:|:---------:|:-----------:|:----:|:----:|:-------:|
> | skip                | 57.3  |   29.7    |    50.9     | 63.9 | 36.5 |  47.7   |
> | skipless                | -  |   -    |    -     | - | - |  -   |
> | skipless + alpha=0 beta=0.5 | 61.2  |   28.2    |    52.2     | 62.7 | 36.5 |  48.2   |
>
> Without our initialization, the standard skipless Transformer diverges immediately. With our method, it achieves comparable performance with the residual baseline. Crucially, while prior work [1] can train skipless Transformers, it requires **5$\times$ more training steps** to match residual performance. In contrast, our method achieves this at the same training speed (1$\times$ steps) as the baseline, demonstrating **significantly superior efficiency**.
>
> - **[1]** He et al., *Deep Transformers Without Shortcuts: Modifying Self-Attention for Faithful Signal Propagation*, ICLR 2023.

---

> ### Author Response · Authors · 2025-11-21
> **Response to Reviewer rJvD**
>
> > **W3:The paper only explores which type of initialization can lead to better conditioning of the network Jacobian, but it does not essentially guarantee better model performance.**
>
> **Response:** The main motivation of our work is that skip connections substantially reduce a model’s effective depth. However, removing them naively introduces severe ill-conditioning in the network Jacobian, which in turn makes optimization highly unstable, as is well known through the Neural Tangent Kernel (NTK) theory [1, 2, 3].
>
> Our initialization is specifically designed to address this challenge by improving the conditioning of skipless Transformers, allowing them to train stably and fully leverage their increased effective depth. This improved conditioning leads to better optimization dynamics and, ultimately, stronger performance in both supervised and self-supervised settings. While conditioning is not itself an evaluation metric, it has been shown in (reference) via the NTK that establishing a well-conditioned Jacobian is a necessary prerequisite for models to converge well and hence for skipless architectures to function effectively, an observation that is consistently supported by our empirical results.
>
> - [1] Jacot et al., Neural Tangent Kernel: Convergence and Generalization in Neural Networks, NeurIPS 2018.
>
> - [2] Yang, Tensor Programs II: Neural Tangent Kernel for Any Architecture, arXiv:2006.14548 2020.
>
> - [3] Liu et al., Loss landscapes and optimization in over-parameterized non-linear systems and neural networks, Applied and Computational Harmonic Analysis 2022.
>
>
> > **Q1: Why does an abrupt inflection point appear in Figure 1? The loss is obviously not smooth**
>
> **Response:** We thank the reviewer for the question. The abrupt inflection in Figure 1 is not due to instability in skipless training but rather to the choice of hyperparameters. For fairness, we used the same hyperparameters as AdamW and did not perform SOAP-specific tuning. Since optimizers typically require task-dependent tuning, these settings are not necessarily optimal for SOAP and can lead to minor irregularities in the loss trajectory.
>
> To clarify this, we provide the full DINO pretraining loss curves in Appendix A.6. Empirically, we find that under our initialization, SOAP exhibits a smooth loss curve, and AdamW appears even smoother. This supports the view that the inflection point arises from hyperparameter mismatch rather than any inherent instability introduced by skipless training.
>
> **We thank you again for reviewing our work. Please let us know if we have misunderstood any of your comments or if you have any follow-up questions. We would be happy to provide further clarification.**
>
> **Best regards,**
>
> **Authors**

---

### Official Review · Reviewer_RZmz · 2025-11-01

**Soundness:** 2
**Presentation:** 2
**Contribution:** 2
**Rating:** 2
**Confidence:** 2

**Summary:**

This paper investigates training ViTs without residual connections. The authors provide a Jacobian-based theoretical analysis showing that skip connections improve network conditioning, and propose a principled initialization strategy that enables stable training of skipless transformers without architectural modifications. The key contributions include: 1. theoretical analysis of transformer Jacobian conditioning; 2. a novel initialization scheme; and 3. empirical validation showing skipless ViTs can match residual models in supervised learning and outperform them in dense prediction tasks.

**Strengths:**

1. Principled Theoretical Foundation: The authors provide some mathematical insights on the Jacobian analysis, which provides clear intuition for why skip connections help optimization.
2. This approach requires no architectural modifications.
3. The authors provide thoughtful experiments.

**Weaknesses:**

1. The restriction to Vision Transformers is a significant limitation.
2. The improvements are minor.

**Questions:**

1. Can you provide more experiments on LLM?
2. Can you ablate the specific properties of SOAP that enable skipless training?
3. Do the benefits hold at larger scales?

---

> ### Author Response · Authors · 2025-11-21
> **Response to Reviewer RZmz**
>
> Dear Reviewer RZmz,
>
> We sincerely appreciate your insightful and constructive feedback, which has been instrumental in improving our work. Please find our detailed point-by-point responses below, along with the corresponding revisions in the manuscript (highlighted in blue).
>
> >**W1: The restriction to Vision Transformers is a significant limitation.**
>
> **Response:** We appreciate the opportunity to clarify the scope and contributions of our work. Skip connections were originally popularized in computer vision through ResNets, making the vision domain a natural testbed for examining how skipless models form hierarchical representations and for understanding how the community might successfully train transformers without residual pathways. Moreover, Vision Transformers are far from a niche architecture: they underpin modern vision foundation models (e.g., SAM, DINO) and serve as the visual encoders in leading multimodal LLMs (e.g., GPT-4o, Gemini, LLaVA). Studying skipless architectures in this setting is therefore highly relevant to current deep learning practice.
>
> That said, we have taken the time to demonstrate broader applicability. To this end, we extended our experiments beyond vision and evaluated our method on language models as well. Specifically, we pretrained a 110M-parameter Transformer on the C4 dataset for 20k iterations using AdamW, and assessed zero-shot performance on five common-sense reasoning tasks [2]. This additional evaluation shows that our approach is not restricted to vision and holds promise for Transformer architectures more broadly.
>
> To further demonstrate that our method is **not limited to vision**, we extended our evaluation to Language Transformers. We pretrained a ~110M parameter model on the C4 dataset for 20k steps (using AdamW) and evaluated zero-shot performance on five common-sense reasoning tasks[2].
>
> | Architecture                 | BoolQ | HellaSwag | Winogrande | PIQA | SIQA | Average |
> |-----------------------------|:-----:|:---------:|:-----------:|:----:|:----:|:-------:|
> | skip                | 57.3  |   29.7    |    50.9     | 63.9 | 36.5 |  47.7   |
> | skipless                | -  |   -    |    -     | - | - |  -   |
> | skipless + alpha=0 beta=0.5 | 61.2  |   28.2    |    52.2     | 62.7 | 36.5 |  48.2   |
>
> Without our initialization, the standard skipless Transformer diverges immediately. With our method, it achieves comparable performance with the residual baseline. Crucially, while prior work [2] can train skipless Transformers, it requires **5$\times$ more training steps** to match residual performance. In contrast, our method achieves this at the same training speed (1$\times$ steps) as the baseline, demonstrating **significantly superior efficiency**. We have now included this analysis on LLMs in a new section in the appendix. Please see Section A.8.
>
>
> [1] Lecun et al., Deep Learning, Nature 2015.
>
> [2] He et al., Deep Transformers Without Shortcuts: Modifying Self-Attention for Faithful Signal Propagation, ICLR 2023.

---

> ### Author Response · Authors · 2025-11-21
> **Response to Reviewer RZmz**
>
> > **W2: The improvements are minor.**
>
> We are very confused with this comment from the reviewer. Our empirical results demonstrate substantial gains across multiple distinct tasks:
>
> - **Supervised classification (ImageNet-1k):**
>       Skipless ViT outperforms vanilla ViT by **+0.7%** in top-1 classification accuracy at the same training speed with the second-order optimizer.
>
> - **Semantic segmentation (linear probing with pretrained DINO ViT):**
>   - Average improvement of **+2.9 mIoU** using single-scale features across **VOC12, COCO-Stuff, ADE20K**.
>   - Average improvement of **+2.0 mIoU** using multi-scale features.
>   - Importantly, these results use **the same pretrained skipless ViT checkpoint**, trained for **200 epochs**, whereas the baseline requires **300 epochs**.  This makes our method **33.3% faster** with the AdamW optimizer, while still achieving stronger performance.
>
>  - **Object discovery (with pretrained DINO ViT):**
>       - VOC12: **+21.7%** (54.0% vs. 32.3%)
>       - COCO20k: **+17.3%** (38.5% vs. 21.2%)
>       - Again, these results rely on **the same 200-epoch skipless checkpoint**, while baselines require **300 epochs**.
>
> These improvements are **far from minor**. They are consistent across multiple benchmarks, and demonstrate clear advantages of the skipless design over the vanilla ViT.

---

> ### Author Response · Authors · 2025-11-21
> **Response to Reviewer RZmz**
>
> > **Q1: Can you provide more experiments on LLM?**
>
> **Response:**  As requested by the reviewer, we have provided more experiments on LLMs (shown above). In each case our methodology shows clear performance gains. We have included this in the appendix of the paper in section A.8.
>
> > **Q2: Can you ablate the specific properties of SOAP that enable skipless training?**
>
> **Response:** We understand the reviewer’s question as asking which aspect of SOAP facilitates training skipless models, and whether the optimizer alone is sufficient. Our findings indicate that SOAP contributes through its second order preconditioner that is used to scale the gradients, which in turn helps mitigate ill-conditioned updates and can accelerate convergence in general. In supervised settings, this sometimes allows skipless models to converge. However, their final performance remains noticeably weaker, and the best results are achieved only when SOAP is combined with our proposed initialization.
>
> The picture changes significantly in the DINO pretraining setup. There, skipless models trained with SOAP still diverge unless they are initialized with our method. This indicates that SOAP’s second-order preconditioning, while beneficial, does not resolve the severe conditioning issues introduced by removing skip connections. Stable optimization is achieved only once our initialization improves the conditioning of the network Jacobian and this can be seen as a result of the fact that the network Jacobian, via Neural Tangent Kernel (NTK) theory [1, 2, 3], plays a role in the convergence to a minimum.
>
> In summary, SOAP does not contain a special mechanism uniquely enabling skipless training. Instead, it offers general optimization benefits, while the key factor that makes skipless training feasible is our initialization, which addresses the underlying conditioning bottleneck that neither AdamW nor SOAP can overcome on their own.
>
> - [1] Jacot et al., Neural Tangent Kernel: Convergence and Generalization in Neural Networks, NeurIPS 2018.
>
> - [2] Yang, Tensor Programs II: Neural Tangent Kernel for Any Architecture, arXiv:2006.14548 2020.
>
> - [3] Liu et al., Loss landscapes and optimization in over-parameterized non-linear systems and neural networks, Applied and Computational Harmonic Analysis 2022.
>
>
> > **Q3: Do the benefits hold at larger scales?**
>
> **Response:** We thank the reivewer for their question. Scaling to very large models (e.g., ViT-Large or ViT-Huge) requires not only substantially larger architectures but also much larger training datasets (such as ImageNet-22K) and significantly greater computational resources. These experiments are unfortunately beyond our current computational budget. We have enhanced our discussion section to a discussion and limitations section, see Section 7, that adds this as a limitation.
>
> Nonetheless, our results already show that the proposed initialization is effective across both supervised and self-supervised regimes. Quantitatively, it stabilizes optimization and consistently outperforms the skip-connection baselines. Qualitatively, skipless ViTs learn more coherent, semantically meaningful hierarchical representations. We believe that this is a novel contribution for the community as it provides a new methodology, via the lens of conditioning, for training skipless transformers in vision applications.
>
> **We thank you again for reviewing our work. Please let us know if we have misunderstood any of your comments or if you have any follow-up questions. We would be happy to provide further clarification.**
>
> **Best regards,**
>
> **Authors**

---

> ### Comment · Reviewer_RZmz · 2025-11-24
> **Official Comment**
>
> Thanks for authors' reply and all other reviewer's comments.
>
> After reading all reply and comments, I still decide to keep my original score. To me, this paper is not good enough to be accepted for ICLR.

---

### Official Review · Reviewer_WDvd · 2025-11-06

**Soundness:** 2
**Presentation:** 3
**Contribution:** 3
**Rating:** 6
**Confidence:** 3

**Summary:**

This paper reformulates the problem of stabilizing training in residual-free transformers as minimizing the condition number of the network Jacobian. It proposes a principled parameter initialization method that improves the conditioning of the self-attention layer. The analysis shows that the Jacobian’s conditioning is dominated by two key matrix products: $W^Q W^{K\top}$ and $W^V W^O$ and that initializing $W^V W^O$ to be scaled-orthonormal and enforcing diagonal dominance in $W^Q W^{K\top}$ can significantly improve conditioning of self-attention layer Jacobian and thus enhance training stability.
Experiments on both supervised learning (ImageNet classification with ViT-Base) and self-supervised learning (DINO feature pretraining with ViT-Small) demonstrate that this initialization enables stable optimization of skipless transformers. The method is compatible well with SOAP optimizer and achieves comparable or better performance than residual-based counterparts. Visualization of the final-layer features  shows that skipless models trained with the proposed initialization yield clearer class boundaries.

**Strengths:**

1. This paper has clear writing and is easy to follow.
2. This paper introduces a novel principled initialization method that is effective on alleviating  instability in residual-free transformer training.
3. This paper conducted experiments on multiple tasks including image-classification and DINO feature pertaining.

**Weaknesses:**

1. **Limited theoretical robustness beyond initialization**. While the principled initialization helps improve conditioning at the beginning of training, but the paper provides no guarantee that good conditioning persists as parameters evolve during training. In contrast, skip connections offer a structural and long-term stabilizer independent of parameter drift.
2. **Simplified input distribution assumption**. The theoretical derivation(Appendix A4.1) assumes token embeddings are drawn from  a normal distribution with mean 0 and identity covariance, which deviates from the empirical distribution after LayerNorm or embedding layers where the tokens are correlated across dimensions and not centered at 0.
3. **Indirect evidence for hierarchical representation**. While visualizations suggest clearer feature boundaries and clean feature maps, no direct qualitative and quantitative evidence is provided to show skipless ViT yields more hierarchical or compositional features than residual ones. Given it’s considered as one strong benefit of skipless ViT. More evidence is supposed to be provided.
4. **Limited model scale diversity**. The experimental validation involves only ViT-Base (supervised) and ViT-Small (self-supervised). Without testing larger models (e.g., ViT-Large or ViT-Huge) and different model size on the same task, it remains unclear whether the initialization scales effectively with increasing parameter count.
5. **Material Missing**. In 5.2, Appendix E is mentioned but is missing in provided materials.

**Questions:**

1. **Optimizer inconsistency: SOAP > AdamW in supervised, but Adam > SOAP in self-supervised**. In the supervised ImageNet setting with ViT-Base, skipless+init trained with a second-order method (SOAP) reaches parity/slight gains vs residual baselines , whereas in DINO pretraining with ViT-Small the best results appear under Adam rather than SOAP (Fig. 2 in the paper). Please analyze why SOAP’s advantage does not carry over to DINO.
2. **Limited theoretical robustness beyond initialization.** Could you track $\kappa(J)$, per-layer $\kappa(K_\ell)$ over epochs for both skipless+init and residual baselines?
3. **More evidence for hierarchical/compositional representations**. Author shows clearer class boundaries from late-layer features in skipless+init models and gains on dense probing/object discovery  , but direct, layer-wise evidence of hierarchy/compositionality is limited. Please provide more evidence for hierarchy/compositionality.
4. **More model scale diversity**. Could you provide the results of ViT-Large and ViT-huge to evaluate scalability of proposed method. (Consider demands on computing resources, it's listed as optional. )
5. **Is minimizing the Jacobian condition number sufficient for stability?** Skipless+init converges to accuracy similar to residual models, but residual training shows smoother loss curves(Figure 1), suggesting potential landscape differences not captured by condition-number alone. If so, will such difference cause instability? (Although this analysis would be valuable and necessary for complete understanding, it falls beyond the scope of this paper. Therefore, it is listed as an optional question.)

 Authors’ rebuttal adequately addressing the first three points is necessary for me to raise the final score.

---

> ### Author Response · Authors · 2025-11-21
> **Response to Reviewer WDvd**
>
> Dear Reviewer WDvd,
>
> We sincerely appreciate your insightful and constructive feedback, which has been instrumental in improving our work. Please find our detailed point-by-point responses below, along with the corresponding revisions in the manuscript (highlighted in blue).
>
> > **W1: Limited theoretical robustness beyond initialization. While the principled initialization helps improve conditioning at the beginning of training, but the paper provides no guarantee that good conditioning persists as parameters evolve during training. In contrast, skip connections offer a structural and long-term stabilizer independent of parameter drift.**
>
> **Response:** We thank the reviewer for raising this thoughtful point. Our initialization scheme is designed to introduce a favorable conditioning bias at the beginning of training, giving the optimizer a more stable starting landscape. Although it is reasonable to question whether this advantage persists throughout the full optimization trajectory, our experiments show that this early-stage improvement is already sufficient to yield consistent performance gains.
>
> Importantly, a well-conditioned Jacobian at initialization plays a central role in stabilizing optimization. Across all our experiments, this principled initialization makes skipless Transformers reliably trainable. We also find empirically that the improved conditioning is largely maintained as training progresses, supporting stable convergence without the need for residual connections.
>
> While residual connections provide strong long-term stability, they also shorten the network’s effective depth. Our goal is to remove the optimization barriers that make skipless Transformers difficult to train, while preserving their full depth and expressive capacity. The proposed initialization offers a viable alternative to residual connections by combining stable early-stage optimization with the architectural benefits of maintaining full depth.

---

> ### Author Response · Authors · 2025-11-21
> **Response to Reviewer WDvd**
>
> > **W2:Simplified input distribution assumption. The theoretical derivation(Appendix A4.1) assumes token embeddings are drawn from a normal distribution with mean 0 and identity covariance, which deviates from the empirical distribution after LayerNorm or embedding layers where the tokens are correlated across dimensions and not centered at 0.**
>
> **Response:** We acknowledge that real-world token distributions do not always follow a Gaussian distribution. However, this assumption enables us to derive clean theoretical insights that directly motivate the architectural choices we propose for training Transformers without skip connections. While the assumption is idealized, it plays the same role as in several prior theoretical works on Transformers [1,2,3,4], and we view it as a standard simplification that facilitates meaningful analysis rather than a claim about the exact nature of all token distributions in practice. Importantly, we also empirically examined this assumption in the revised paper in Figure 6, Appendix A.4.2, where we visualize the distribution of token embeddings after LayerNorm and found that it closely approximates a zero-mean, unit-variance Gaussian. Taken together, the assumption is both theoretically useful and empirically reasonable, and it ultimately guides us toward a novel and practical methodology for training Transformers without skip connections.
>
> - [1] Xiong et al., On Layer Normalization in the Transformer Architecture, ICML 2020.
> - [2] Takase et al., Spike No More: Stabilizing the Pre-training of Large Language Models, COLM 2025.
> - [3] Li et al., MIX-LN: Unleashing the Power of Deep Layers by Combining Pre-LN and Post-LN, ICLR 2025.
> - [4] Ramapuram et al., Theory, Analysis, and Best Practices for Sigmoid Self-Attention, ICLR 2025.

---

> ### Author Response · Authors · 2025-11-21
> **Response to Reviewer WDvd**
>
> > **W3: Indirect evidence for hierarchical representation. While visualizations suggest clearer feature boundaries and clean feature maps, no direct qualitative and quantitative evidence is provided to show skipless ViT yields more hierarchical or compositional features than residual ones. Given it’s considered as one strong benefit of skipless ViT. More evidence is supposed to be provided.**
>
> **Response:** To illustrate the emergence of hierarchical representations, we focus on images containing a clearly identifiable dominant object. In these cases, skipless ViTs tend to exhibit a coherent progression: early layers highlight meaningful subparts of the object, while deeper layers capture the full object structure. In contrast, residual baselines show noticeably weaker or blurrier patterns along this hierarchy. For clarity, we have provided the corresponding visualizations in the revised paper in Appendix A.5.
>
> > **W4:Limited model scale diversity. The experimental validation involves only ViT-Base (supervised) and ViT-Small (self-supervised). Without testing larger models (e.g., ViT-Large or ViT-Huge) and different model size on the same task, it remains unclear whether the initialization scales effectively with increasing parameter count.**
>
> **Response:** We thank the reviewer for their comment. Scaling to very large models (e.g., ViT-Large or ViT-Huge) requires not only substantially larger architectures but also much larger training datasets (such as ImageNet-22K) and significantly greater computational resources.
> These large-scale experiments are unfortunately beyond our current computational budget. Nevertheless, our results already show that the proposed initialization is effective across both supervised ViT-Base and self-supervised ViT-Small settings. While exploring scaling laws for skipless Transformers is an exciting and valuable future direction, it is complementary to, rather than essential for, the core contributions of this work. Given that our methodology is one of the first that shows how to train ViTs without skip connections in the setting of applications such as DINO, we believe it poses a useful contribution for the community and will lead to further research on the training of skipless architectures. We have enhanced our discussion section to a discussion and limitations section, see Section 7, that adds this as a limitation.
>
> > **W5: Material Missing. In 5.2, Appendix E is mentioned but is missing in provided materials.**
>
> **Response:**  We thank the reviewer for bringing this to our attention. We have corrected this in revised version of our paper.

---

> ### Author Response · Authors · 2025-11-21
> **Response to Reviewer WDvd**
>
> > **Q1: Optimizer inconsistency: SOAP > AdamW in supervised, but Adam > SOAP in self-supervised. In the supervised ImageNet setting with ViT-Base, skipless+init trained with a second-order method (SOAP) reaches parity/slight gains vs residual baselines , whereas in DINO pretraining with ViT-Small the best results appear under Adam rather than SOAP (Fig. 2 in the paper). Please analyze why SOAP’s advantage does not carry over to DINO.**
>
> We appreciate the reviewer’s observation regarding the performance differences between SOAP and Adam. In our view, this variability reflects a broader, well-known challenge in optimization rather than a limitation of our approach. Prior work has consistently shown that optimizer performance is highly problem-dependent [1,2], and that there is no universally superior optimizer across tasks. More recent studies [3] further demonstrate that different optimizers induce distinct inductive biases, often leading models to converge to qualitatively different solutions. Given this, it is natural that Adam and SOAP behave differently depending on the loss objective, and, since no existing theory can predict the optimal optimizer–loss pairing, the community typically relies on empirical selection.
>
> Within this context, our contribution is to show that skipless ViTs can train as efficiently as, and sometimes more efficiently than, residual-based ViTs when paired with a suitable optimizer. This is noteworthy because the only previous attempt at training skipless Transformers [4] required roughly 5× more training steps to match residual-based performance, even when using a stronger second-order method.
>
> Finally, we believe our results highlight the importance of exploring optimizers beyond the standard Adam/AdamW family, particularly in vision tasks where alternative optimization strategies remain relatively underexplored. Since different loss functions interact with optimizers in fundamentally different ways, our findings suggest a promising direction for future investigation into the optimizer–architecture interplay in Transformer training.
>
> - [1] Schmidt et al., Descending through a Crowded Valley — Benchmarking Deep Learning Optimizers, ICML 2021.
> https://proceedings.mlr.press/v139/schmidt21a.html
>
> - [2] Schneider et al., DEEPOBS: A Deep Learning Optimizer Benchmark Suite, ICLR 2019.
> https://openreview.net/forum?id=rJg6ssC5Y7
>
> - [3] Pascanu et al., Optimizers Qualitatively Alter Solutions And We Should Leverage This, 2025.
> https://arxiv.org/pdf/2507.12224
>
> - [4] He et al., Deep Transformers Without Shortcuts: Modifying Self-Attention for Faithful Signal Propagation, ICLR 2023.
> https://openreview.net/pdf?id=NPrsUQgMjKK

---

> ### Author Response · Authors · 2025-11-21
> **Response to Reviewer WDvd**
>
> > **Q2: Limited theoretical robustness beyond initialization. Could you track $\kappa(J)$, per-layer $\kappa(K_\ell)$ over epochs for both skipless+init and residual baselines?**
>
> **Response:** We thank the reviewer for suggesting this analysis. As requested, we tracked the condition numbers of the full Jacobian ($\kappa(\mathbf{JJ}^\top)$) and per-layer Jacobians during DINO ViT-Small training.
>
> 1. Full Jacobian $\kappa(\mathbf{JJ}^\top)$
>     As shown in the table below, the standard skipless baseline diverges, leading to numerical overflow ($\infty$). In contrast, our skipless + init maintains a condition number comparable in magnitude to the skip (residual) baseline throughout the entire training process. This confirms that our initialization prevents the optimization difficulty that typically happens in skipless architectures.
>
> |                 | Epoch 20 | Epoch 60 | Epoch 120 | Epoch 180 | Epoch 240 | Epoch 300 |
> |:----------------|:-------:|:--------:|:---------:|:---------:|:---------:|:---------:|
> | skip            |   127   |    72    |    82     |     93    |     113    |    144    |
> | skipless        | $\infty$|$\infty$  |$\infty$   |  $\infty$ | $\infty$   | $\infty$  |
> | skipless + init |   195   |    340   |    261    |    160    |    176     |    207    |
>
> 2. Layer-wise ($\kappa(\mathbf{K_\ell K_\ell^\top})$ for skipless ViT and $\kappa(\mathbf{(K_\ell+I) (K_\ell+I)^\top})$ for skip ViT). We further analyzed the condition numbers of the Jacobian kernels across depths.
>
> |     Epoch = 20   | layer 0 | layer 4 | layer 7 | layer 11 |
> |:----------------|:-------:|:-------:|:-------:|:--------:|
> | skip            |   1.05   |    1.49    |    1.35     |    1.04     |
> | skipless        | $\infty$  |   $\infty$     |    $\infty$      |   $\infty$      |
> | skipless + init |   10  |    2.7    |    7.5     |    12.5     |
>
> |     Epoch = 100   | layer 0 | layer 4 | layer 7 | layer 11 |
> |:----------------|:-------:|:-------:|:-------:|:--------:|
> | skip            |   1.06   |    1.54    |    1.25     |    1.25     |
> | skipless        | $\infty$ | $\infty$| $\infty$|$\infty$ |
> | skipless + init |   6.25  |    3.7    |    8.54    |    19.2     |
>
>
> |     Epoch = 200   | layer 0 | layer 4 | layer 7 | layer 11 |
> |:----------------|:-------:|:-------:|:-------:|:--------:|
> | skip            |   1.03   |    1.57    |    1.21     |    1.37     |
> | skipless        | $\infty$ | $\infty$| $\infty$|$\infty$ |
> | skipless + init |   6.25  |    3.84    |    11.1    |    21.7     |
>
> |     Epoch = 300   | layer 0 | layer 4 | layer 7 | layer 11 |
> |:----------------|:-------:|:-------:|:-------:|:--------:|
> | skip            |   1.02   |    1.38    |    1.28     |    1.51     |
> | skipless        | $\infty$ | $\infty$| $\infty$|$\infty$ |
> | skipless + init |   3.44  |    3.57    |    14.3    |    20.2     |
>
> These results empirically demonstrate that our theoretical derivation holds beyond initialization, ensuring robust trainability for deep skipless networks.
> We also include this in the revised manuscript (Table 5 and 6 in Appendix A.7)

---

> ### Author Response · Authors · 2025-11-21
> **Response to Reviewer WDvd**
>
> > **Q3: More evidence for hierarchical/compositional representations. Author shows clearer class boundaries from late-layer features in skipless+init models and gains on dense probing/object discovery , but direct, layer-wise evidence of hierarchy/compositionality is limited. Please provide more evidence for hierarchy/compositionality.**
>
> **Response:** See our response to weakness 3. We have added such visualizations in Appendix A.5.
>
> > **Q4: More model scale diversity. Could you provide the results of ViT-Large and ViT-huge to evaluate scalability of proposed method. (Consider demands on computing resources, it's listed as optional. )**
>
> **Response:** We thank the reviewer for the question. Unfortunately, we do not currently have access to the computational resources required to run experiments at that scale. Nevertheless, we believe the contributions of our work remain significant. To the best of our knowledge, this is the first demonstration that a skipless Transformer can be successfully trained for applications involving Vision Transformers. More broadly, we expect that the training insights introduced in this paper will help guide future efforts towards developing more efficient methodologies for scaling skipless Transformers to larger, billion-parameter regimes.
>
> > **Q5: Is minimizing the Jacobian condition number sufficient for stability? Skipless+init converges to accuracy similar to residual models, but residual training shows smoother loss curves(Figure 1), suggesting potential landscape differences not captured by condition-number alone. If so, will such difference cause instability? (Although this analysis would be valuable and necessary for complete understanding, it falls beyond the scope of this paper. Therefore, it is listed as an optional question.)
>
> **Response:** We thank the reviewer for this insightful question. Minimizing the condition number of the Jacobian improves optimization stability, as motivated by Neural Tangent Kernel (NTK) theory [1, 2, 3], where better-conditioned Jacobians correspond to more stable convergence dynamics. However, the smoothness of the loss landscape is governed by the Hessian of the loss. The Jacobian condition number does not directly characterize this smoothness, and obtaining such information would require computing or approximating the Hessian, an operation that is prohibitively expensive for large-scale models such as Transformers. For this reason, we focus on Jacobian conditioning, which provides a tractable and theoretically meaningful proxy for understanding and improving optimization behavior.
> Exploring interventions that both directly smooth the loss landscape and condition the Jacobian, particularly without relying on skip connections, is a compelling direction for future work but falls outside the scope of the present study research.
>
>
> - [1] Jacot et al., Neural Tangent Kernel: Convergence and Generalization in Neural Networks, NeurIPS 2018.
>
> - [2] Yang, Tensor Programs II: Neural Tangent Kernel for Any Architecture, arXiv:2006.14548 2020.
>
> - [3] Liu et al., Loss landscapes and optimization in over-parameterized non-linear systems and neural networks, Applied and Computational Harmonic Analysis 2022.
>
>
> **We thank you again for reviewing our work. Please let us know if we have misunderstood any of your comments or if you have any follow-up questions. We would be happy to provide further clarification.**
>
> **Best regards,**
>
> **Authors**

---

### Meta-Review · Area_Chair_ERBU · 2026-01-05

**Summary:**

The reviewers expressed a mix of interest and skepticism regarding the possibility of training Transformers without residual connections. The primary concerns that informed the initial assessments included the perceived limitation of the approach to Vision Transformers (ViTs) , the long-term stability of a method based purely on initialization rather than structural stabilizers , and whether the reported performance gains were significant enough to warrant a departure from standard residual architectures. Additionally, reviewers requested more direct evidence of the "hierarchical representations" claimed to be a benefit of skipless designs and questioned the scalability of the method to larger models or different domains like Large Language Models (LLMs)

**Reviewer Concerns:**

authors’ rebuttal successfully addressed the most critical technical and empirical challenges raised during the review process.

**Stability Beyond Initialization**: To address Reviewer WDvd’s concern that the initialization might only help at the start of training , the authors provided empirical tracking of the Jacobian condition number throughout 300 epochs of training. This data demonstrated that the stable conditioning persists over the full optimization trajectory.

**Generalization to LLMs**: In response to Reviewer RZmz’s critique regarding the restriction to vision , the authors conducted new experiments pretraining a 110M-parameter Transformer on the C4 dataset. The results showed that the skipless model achieved zero-shot performance comparable to residual baselines while maintaining 1x training speed, a significant improvement over prior work.

**Hierarchical Evidence**: The authors provided new visualizations in the revised Appendix A.5 showing a coherent progression from sub-parts to full objects in skipless models, which was noticeably blurrier in residual baselines.

**Empirical Significance**: The authors clarified that the gains are substantial: a +0.7% top-1 accuracy on ImageNet and up to +2.9 mIoU in semantic segmentation, achieved with 33.3% faster training compared to the required 300-epoch baseline.

The primary outstanding concern is the scalability to billion-parameter models (e.g., ViT-Large/Huge, or other *real* large language models), which the authors could not address due to computational constraints. However, they have acknowledged this as a formal limitation in the revised manuscript.

**Reviewer Scores:**

Reviewer **WDvd** (Initial: 6): Given that the authors specifically addressed the "first three points" (stability tracking, distribution assumptions, and hierarchy evidence) which the reviewer cited as necessary for a score increase, this reviewer would likely have moved to a 7 or 8.

Reviewer **RZmz** (Initial: 2): This reviewer showed a lack of engagement, ignoring the factual rebuttal regarding performance gains and the new experiments they specifically requested.

Reviewer **rJvD** (Initial: 6): With the added evidence of hierarchical representations and the successful application of the method to general Transformer architectures, this reviewer would likely have moved to a 7.

---

### Decision · Program_Chairs · 2026-01-26

Accept (Poster)